# A Survey of Frontiers in LLM Reasoning: Inference Scaling, Learning to Reason, and Agentic Systems

Zixuan Ke[*]                                                    zixuan.ke@salesforce.com

Fangkai Jiao[◇,‡]                                              jiaofangkai@hotmail.com

Yifei Ming[*]                                                   yifei.ming@salesforce.com

Xuan-Phi Nguyen[*]                                          xnguyen@salesforce.com

Austin Xu[*]                                                    austin.xu@salesforce.com

Do Xuan Long[†,‡]                                           xuanlong.do@u.nus.edu

Minzhi Li[†‡]                                                    li.minzhi@u.nus.edu

Chengwei Qin[◇]                                            chengwei003@e.ntu.edu.sg

Peifeng Wang[*]                                           peifeng.wang@salesforce.com

Silvio Savarese[*]                                          ssavarese@salesforce.com

Caiming Xiong[*]                                              cxiong@salesforce.com

Shafiq Joty[*,◇]                                                sjoty@salesforce.com

[*] *Salesforce AI Research*          [†] *National University of Singapore*
[◇] *Nanyang Technological University*          [‡] *I²R, A\*STAR, Singapore*

## Abstract

Reasoning is a fundamental cognitive process that enables logical inference, problem-solving, and decision-making. With the rapid advancement of large language models (LLMs), reasoning has emerged as a key capability that distinguishes advanced AI systems from conventional models that empower chatbots. In this survey, we categorize existing methods along two orthogonal dimensions: (1) *Regimes*, which define the stage at which reasoning is achieved (either at inference time or through dedicated training); and (2) *Architectures*, which determine the components involved in the reasoning process, distinguishing between standalone LLMs and agentic compound systems that incorporate external tools, and multi-agent collaborations. Within each dimension, we analyze two key perspectives: (1) *Input* level, which focuses on techniques that construct high-quality prompts that the LLM condition on; and (2) *Output* level, which methods that refine multiple sampled candidates to enhance reasoning quality. This categorization provides a systematic understanding of the evolving landscape of LLM reasoning, highlighting emerging trends such as the shift from inference-scaling to learning-to-reason (e.g., DeepSeek-R1), and the transition to agentic workflows (e.g., OpenAI Deep Research, Manus Agent). Additionally, we cover a broad spectrum of learning algorithms, from supervised fine-tuning to reinforcement learning such as PPO and GRPO, and the training of reasoners and verifiers. We also examine key designs of agentic workflows, from established patterns like generator-evaluator and LLM debate to recent innovations. Finally, we identify emerging trends, such as domain-specific reasoning systems, and open challenges, such as evaluation and data quality. This survey aims to provide AI researchers and practitioners with a comprehensive foundation for advancing reasoning in LLMs, paving the way for more sophisticated and reliable AI systems.

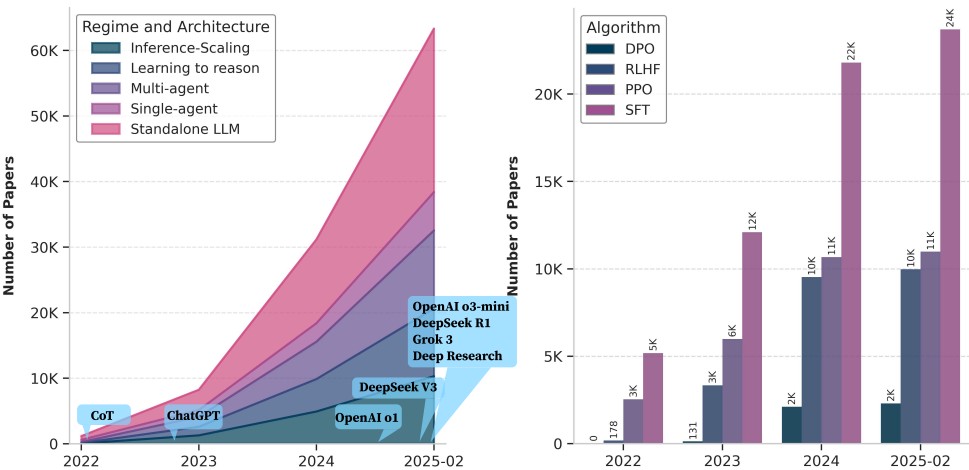

Figure 1: The LLM reasoning surge. We show the cumulative number (in thousands) of papers published from 2022 to 2/2025, based on Semantic Scholar keyword search. Research on reasoning regimes and agent architectures has accelerated notably since the introduction of Chain-of-Thought (CoT) in 2022. This growth is further influenced by other major developments, such as the release of ChatGPT (Ouyang et al., 2022) in 9/2022, and popularity of in-context learning (Brown et al., 2020) as an inference-time optimization method.

# 1 Introduction

Reasoning is the cognitive process of analyzing evidence, constructing arguments, and applying logic to form conclusions or make informed judgments. It is essential to many intellectual pursuits, including decision-making, problem-solving, and critical thinking. The study of reasoning spans multiple disciplines—philosophy (Passmore, 1961), psychology (Wason & JohnsonLaird, 1972), and computer science (Huth & Ryan, 2004)—as it provides insights into how individuals interpret information, evaluate alternatives, and develop sound conclusions using logic.

Recently, large language models (LLMs) have demonstrated a range of emerging abilities, such as in-context learning (Dong et al., 2024), role playing (Shanahan et al., 2023b) and domain adaptation (Ke et al., 2023; 2025a; Ke & Liu, 2023) as they scale, with reasoning becoming one of the most critical capabilities. As shown in Figure 1, this area has rapidly gained research attention, often referred to as *LLM reasoning* or *reasoning language model* (RLM) (Besta et al., 2025). The increasing focus on this topic is understandable, as reasoning capability is: (i) **Challenging**, requiring multi-step processing beyond the token-by-token generative nature of auto-regressive LLMs; (ii) **Fundamental**, as it is a core aspect of intelligence, particularly in planning and strategic decision-making; and, most importantly, (iii) **Promising**, as recent advances in LLMs hint at a viable path forward. Given these factors, reasoning is widely regarded as a prerequisite for more advanced AI systems approaching Artificial General Intelligence (AGI), beyond the conventional AI that aims to closely follow instruction (Duenas & Ruiz, 2024).

Reasoning requires LLMs to go beyond directly producing an answer from a question; instead, they must generate the thinking process (implicitly or explicitly) in the form of 'question → reasoning steps → answer'. It has been shown that scaling pre-training may not be the optimal solution for improving reasoning (Snell et al., 2025; OpenAI, 2025). Instead, one popular approach to achieve this is the well-known chain-of-thought (CoT) prompting (Wei et al., 2022b), which demonstrates that by modifying the prompt (e.g., 'Let us think step by step') or in-context samples, LLMs can elicit a step-by-step reasoning process at test time without additional training. Such intuitive prompting techniques have been shown to substantially improve LLMs' reasoning accuracy (Wei et al., 2022b). Building on this, the ability of LLMs to reason effectively depends on two factors: how and at what stage reasoning is achieved, and what components are involved in the reasoning process. Accordingly, in this survey, we categorize existing research into two orthogonal dimensions: **(1) Regime**, refers to whether reasoning is achieved through inference-time strategies (aka. inference-time

scaling) or through direct learning and adaptation (learning to reason); and **(2) Architecture**, refers to whether reasoning happens within a single, standalone LLM or within an interactive, agentic system.

These two dimensions are orthogonal, meaning different regimes can be applied to the same architecture, and different architectures can operate under the same regime. The intersection of these dimensions allows for a more comprehensive and systematic organization of reasoning techniques, encompassing most approaches studied to date while highlighting key trends, such as the shift from inference scaling to learning-to-reason and from standalone LLMs to agentic systems. Notably, most prior surveys have focused on only one or two of these dimensions, typically inference scaling and standalone LLMs, rarely considering both together (see detailed comparison later). By introducing this categorization, we aim to provide a structured perspective that clarifies the diverse landscape of LLM reasoning and establishes a foundation for future research.

## 1.1 Reasoning Regimes

**Inference scaling**  CoT prompting demonstrates the potential to scale inference-time (test-time) reasoning. It has also been shown that optimal scaling of test-time compute can be more effective than scaling model parameters (Snell et al., 2024), as it improves generalization through enhanced flexibility in prompt and workflow design. Building on this, **inference scaling** techniques have emerged, allowing additional test-time computation before generating an answer. The key idea is that instead of updating the LLM itself, these methods aim to select the best trajectories to improve reasoning.

Several variants of prompting methods (Paranjape et al., 2021; Sanh et al., 2022; Mishra et al., 2022) have been introduced, providing structured prompts to enhance reasoning. Additionally, inference scaling optimizes reasoning through search and planning (Dua et al., 2022; Zhou et al., 2023a; Khot et al., 2023; Suzgun & Kalai, 2024a). One key challenge in search and planning is evaluating the quality of candidate solutions. However, evaluating reasoning quality is inherently difficult, even for humans. Existing approaches can be categorized based on whether they judge the final outcome, i.e., outcome reward models (ORMs) (Hendrycks et al., 2021b), or the reasoning process, i.e., process reward models (PRMs) (Lightman et al., 2024).

One of the most notable milestones in this direction is OpenAI's o1 (09/2024) (OpenAI et al., 2024), which demonstrate the effectiveness of inference-time scaling in complex tasks like mathematics, coding and scientific problem-solving:

> *"We have found that the performance of o1 consistently improves with more reinforcement learning (train-time compute) and with more time spent thinking (test-time compute). The constraints on scaling this approach differ substantially from those of LLM pretraining, and we are continuing to investigate them."* — OpenAI o1 release blog

**Learning-to-reason**  Another approach to unleash the deliberate thinking is updating the LLM through training. Unlike inference scaling, learning-to-reason aims to enhance reasoning capabilities through dedicated training, reducing reliance on costly inference-time computations. However, a key challenge in this regime is the scarcity of training data, as step-by-step human-annotated reasoning trajectories are prohibitively expensive to collect. To address this, research has focused on automatically generating such trajectories and developing effective training strategies to leverage them. For example, supervised fine-tuning with long CoT (Muennighoff et al., 2025) or preference learning with reasoning preference data, with DPO (Rafailov et al., 2023) as a representative approach. More recent approaches even bypass reasoning annotation by using reinforcement learning (RL), with recent work like GRPO (Shao et al., 2024) demonstrating remarkable success in this direction. A significant milestone in this direction is DeepSeek-R1 (01/2025) (DeepSeek-AI et al., 2025), an open-source model that achieves performance comparable to OpenAI's o1 while requiring far fewer computational resources. It further reveals that RL alone is possible to learn the sophisticated behaviors just as the test-time computation increase:

> *"One of the most remarkable aspects of this self-evolution is the emergence of sophisticated behaviors as the test-time computation increases. Behaviors such as reflection—where the model revisits and reevaluates its previous steps—and the exploration of alternative ap-*

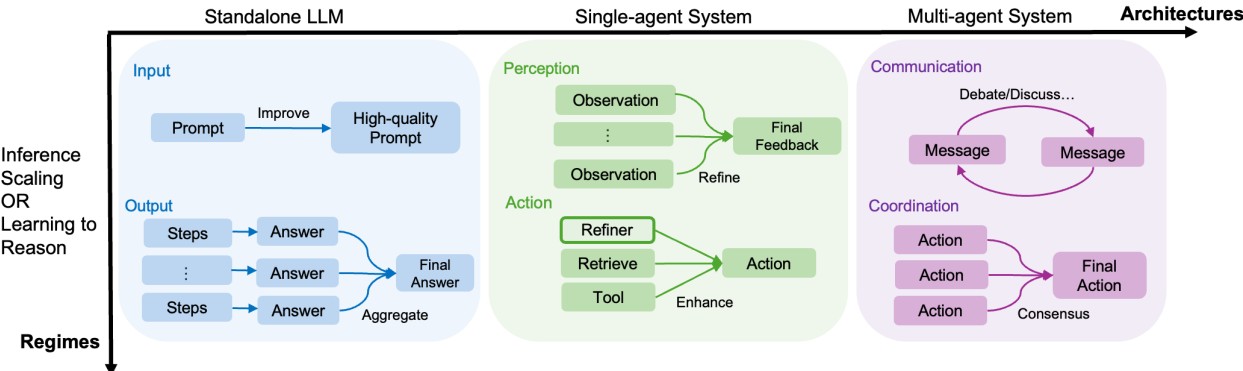

Figure 2: The proposed categorization over regimes, architectures, and unified perspectives in this survey.

> *proaches to problem-solving arise spontaneously. These behaviors are not explicitly programmed but instead emerge as a result of the model's interaction with the reinforcement learning environment."* — DeepSeek-R1 'Aha moment'

## 1.2 Reasoning System Architecture

**Standalone LLM and agentic systems**  Orthogonal to the regimes, studies have explored architectural advancements in LLM reasoning, moving beyond next-token prediction in standalone models to embrace *agentic systems*—AI systems that exhibit interactivity and autonomy to refine reasoning and decision-making. These systems go beyond the challenges of inference scaling or learning to reason; they introduce system-level complexities, such as designing workflows and coordinating potentially conflicting actions.

**Single-Agent and multi-agent systems**  To distinguish agentic systems from standalone LLMs, we adopt the perspective of Kapoor et al. (2024), framing agentic behavior as a spectrum. We categorize these systems into two families: *single-agent* and *multi-agent*. In single-agent systems, a single LLM interacts with tools in its environment to refine reasoning, actions, and perceptions. These tools include external knowledge bases (Ke et al., 2024; Hammane et al., 2024; Sun et al., 2023), verifiers (Wan et al., 2024c; Guan et al., 2025), and practical applications like code interpreters, calendars, and maps (Yu et al., 2023b; Lu et al., 2024a). By leveraging these resources, the LLM iteratively enhances its decision-making and problem-solving capabilities. Recent milestones in single-agent systems, such as Grok 3 Deep Search (02/2025) and OpenAI Deep Research (02/2025), demonstrate how agents interact with the web to significantly improve reasoning, perform tasks like information retrieval, use code interpreters for calculations, and aggregate data from multiple sources.

> *"Deep research independently discovers, reasons about, and consolidates insights from across the web. To accomplish this, it was trained on real-world tasks requiring browser and Python tool use … While o1 demonstrates impressive capabilities in coding, math, and other technical domains, many real-world challenges demand extensive context and information gathering from diverse online sources."* — OpenAI deep research release blog

The second family, multi-agent systems, goes beyond agent-environment interactions by enabling agent-agent communication. Each agent takes on a distinct role and exchanges messages with others. Key challenges include designing effective communication protocols—whether collaborative (Chen et al., 2023c) or adversarial (Liang et al., 2023b)—and coordinating actions to reach consensus on the final action for the environment. A recent example of this potential is Manus, a popular product showcasing the power of multi-agent systems.

### 1.3  Unified Perspectives

Although inference scaling and learning-to-reason take different approaches to improving reasoning, they are inherently connected. Inference scaling focuses on selecting the best reasoning trajectories, while learning-to-reason leverages both good and bad trajectories as training data. To unify these approaches, we categorize reasoning trajectory collection techniques in both regimes based on two key perspectives: **input** and **output**. At the input level, techniques modify or augment prompts to guide the LLM toward desirable reasoning paths. At the output level, the LLM generates multiple candidate responses, which are then evaluated, ranked, or refined. This framework highlights that many inference scaling techniques—such as prompt modification or trajectory search—can be repurposed for trajectory collection in learning-to-reason (as described in Section 3 and Section 5). Moreover, this connection shows that the two approaches are complementary: inference scaling methods can be applied to models trained under learning-to-reason, motivating the development of inference-aware learning-to-reason methods (Section 5.4).

These aspects are also effective across different architectures. Similar to standalone LLMs, we categorize techniques based on input and output perspectives. However, to align with agentic system conventions, we use **perception** as input (to an agent) and **action** as output (of an agent) in single-agent systems. For multi-agent systems, we consider **communication** as input (to a participating agent) and **coordination** as output (of the system). This analogy provides a unified perspective across regimes and architectures, offering a systematic and generalizable framework for analyzing LLM reasoning (see Figure 2).

### 1.4  Goal and Structure of the Survey

The goal of this survey is to provide a comprehensive overview of key algorithmic details and major milestones in LLM reasoning research, particularly since the emergence of Chain-of-Thought (CoT), across both regime and architecture dimensions. We believe this is a timely and valuable contribution to the community, given the clear acceleration in research following CoT's introduction in 2022 (Figure 1). The rapid growth in studies exploring all aspects of LLM reasoning—from regimes and architectures to training algorithms—highlights the increasing importance and utility of reasoning capabilities in advancing the field.

Figure 2 provides an overview of the categorization in this survey, organized along two orthogonal dimensions. Within each architecture, there are two key perspectives to consider. The first perspective is input, or perception, or communication. This concerns how to construct a better prompt, refine the given observations from the environment, or establish protocols for exchanging messages with other agents. The second is output—encompassing action or coordination—which involves aggregating outputs, enhancing actions, or coordinating actions to produce a final result. While the figure illustrates high-level categorizations, the following sections delve into more specific terms. For example, 'input' is discussed in terms of constructing prompts (see e.g., Sections 3.1.1 and 5.1.1), while 'output' relates to optimizing output and collecting high-quality trajectories (e.g., Sections 3.1.2 and 5.1.2).

Figure 3 outlines the structure of this survey. We start with a brief introduction to the background, covering key terminologies, components, regimes, and architectures (Section 2). The subsequent sections explore inference scaling (Section 3), learning algorithms for reasoners and verifiers (Section 4), and learning to reason (Section 5). Within the discussions on inference scaling and learning to reason, we examine three key architectures: Standalone LLMs, Single-Agent systems, and Multi-Agent systems. Finally, Section 6 summarizes key insights and discusses open challenges and future directions.

### 1.5  Comparison to Related Surveys

Reasoning in LLMs has long been a fundamental challenge in the field. Earlier works, such as Huang & Chang (2023), provide a comprehensive overview of the evolution of informal deductive reasoning covering developments *prior to* the emergence of LLM agents and Reasoning Language Models (RLMs). Our work extends this discussion by focusing on LLM agents and RLMs. Qiao et al. (2023b) offer a detailed summary of advancements in LLM reasoning, with a particular emphasis on prompting techniques. In contrast, we offer a broader range of regimes (from inference to training) and architectures (from standalone LLM to multi-agent systems). Readers interested in a formal definition and taxonomy of natural language reasoning—grounded

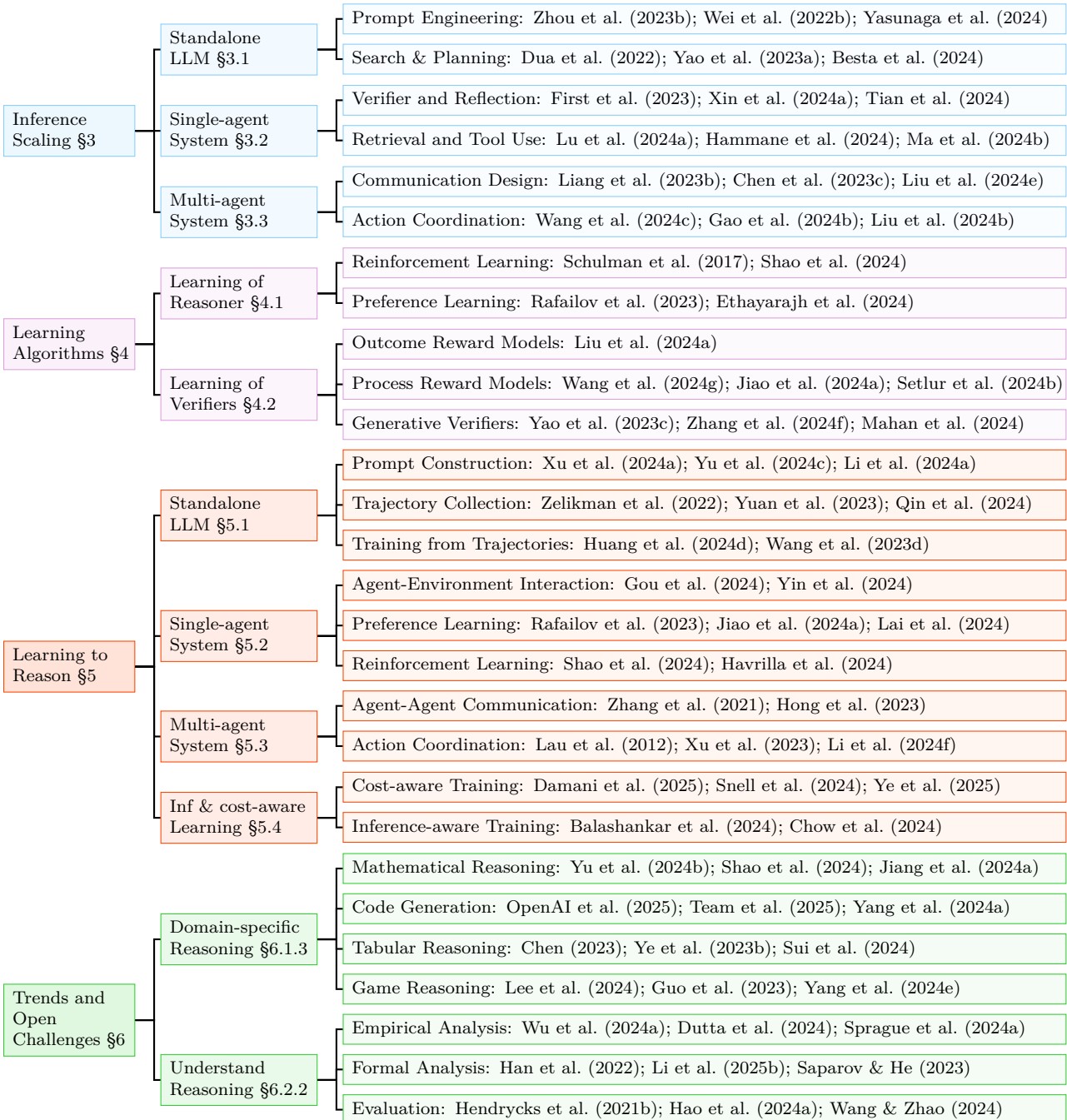

Figure 3: Taxonomy of LLM reasoning research organized in this survey by regimes (inference scaling, learning to reason) and architectures (standalone LLM, single-agent, multi-agent). Each leaf node includes examples from the literature that focus on the corresponding category.

in philosophical foundations—may refer to Yu et al. (2024a), which focuses specifically on this direction and is complementary to our scope.

Improvements in LLM reasoning are closely tied to advancements in a variety of techniques. Dong et al. (2024) present a comprehensive survey on in-context learning (ICL), while Zhou et al. (2024c) explore the interpretation and analysis of ICL from both theoretical and empirical perspectives. In contrast, our work organizes ICL techniques under different regimes—standalone LLMs, single-agent, and multi-agent

systems—highlighting how these techniques evolve and interact within each setting. Recent studies suggest that enhancements in reasoning are often linked to inference scaling. Dong et al. (2024) provide an extensive review of inference-time self-improvement, and Welleck et al. (2024) offer a survey focused on three key themes: token-level generation algorithms, meta-generation algorithms, and efficient generation. Following the release of Reasoning Language Models (RLMs) such as OpenAI's o1 and DeepSeek's R1, there has been a significant increase in research dedicated to learning-to-reason approaches. Zeng et al. (2024) and Xu et al. (2025d) provide thorough surveys on these emerging developments. However, these surveys primarily focus on LLMs, and do not address agentic or multi-agent reasoning settings in depth.

Research on LLM reasoning has predominantly centered on logical and mathematical reasoning. Liu et al. (2025a) offer a comprehensive survey of logical reasoning in LLMs, delving into its theoretical foundations and associated benchmarks. In their position paper, Yang et al. (2024d) underscore the pivotal role of formal mathematical reasoning, showcasing its superiority over traditional NLP-based methods in generating verifiable proofs and automated feedback. Their work outlines progress in theorem proving and auto-formalization while identifying key challenges that remain. While we cover domain-specific reasoning in Section 6.1.3, we refer readers to Liu et al. (2025a) and Yang et al. (2024d) for a more in-depth treatment of these topics.

Reasoning is a critical capability in agentic systems (Pezeshkpour et al., 2024; Masterman et al., 2024). While numerous reviews focus on agent systems (Xi et al., 2023; Kapoor et al., 2024), discussions on reasoning within these systems remain limited. A concurrent work by Besta et al. (2025) introduces a comprehensive and modular framework for RLMs that systematically organizes key components such as reasoning structures, strategies, benchmarks and learning algorithms. However, their work does not delve into agentic and multi-agent LLM systems.[1]

This survey provides a comprehensive overview of major milestones in LLM reasoning research, emphasizing two key dimensions: (1) the evolution of learning schemes—from inference scaling to learning-to-reason approaches—and (2) architectural advancements—from single LLMs to multi-agent systems. These dimensions summarize recent progress and lay the groundwork for future reasoning LLMs and agentic systems. We unify techniques under input and output perspectives, clarifying what must be customized or designed when building reasoning systems. Additionally, we detail essential techniques, including a comparison of the latest learning algorithms (e.g., RL) and an in-depth discussion of refiners and verifiers, which are critical for facilitating reasoning. Given these contributions, our survey is timely, offering AI researchers up-to-date insights into the field. We anticipate further research along these dimensions, such as agent-human regimes (Liang et al., 2024) and automated workflow design architectures (Hu et al., 2025; Zhang et al., 2024c; Zhou et al., 2025a).

## 2  Background

In this section, we introduce foundational concepts that will be utilized throughout the paper.

### 2.1  Problem Formulation

LLM reasoning is often formulated within the Markov Decision Process (MDP) framework (Bellman, 1958), treating reasoning as a sequential decision-making process. While many of the terminologies in LLM reasoning originate from the AI agent and reinforcement learning (RL) literature (Russell & Norvig, 2010), their meaning in LLM reasoning can sometimes differ to suit the nature of LLM-based reasoning.

**Reasoning step and thought**   The definition of what makes a reasoning step can vary depending on the specific inference or learning algorithm used, and it often depends on the granularity at which rewards (or feedback) are considered. Generally, a reasoning step can be expressed as a sequence of tokens $a_t = (x_{t_1}, \ldots, x_{t_K})$, where $x_{t_k}$ is the $k$-th token at inference step $t$. Typically, $a_t$ represents a coherent step in reasoning (Lightman et al., 2024), such as a logical deduction or an intermediate conclusion. However, in extreme cases, a reasoning step can be the entire response (Zhang et al., 2024b; DeepSeek-AI et al., 2025)

---

[1]To avoid redundancy with existing literature, we do not include an analysis of reasoning benchmarks in this survey. For a detailed discussion of benchmarks, we direct readers to Xu et al. (2025d); Besta et al. (2025).

| Symbol | Name/terminology | Explanation |
|---|---|---|
| $a_t$ | Action/response | The reasoning step or action taken at time step $t$ , where $t \in \{1, 2, ..., T\}$ |
| $s_t$ | State/context | $s_t := (q, a_1, ..., a_{t-1})$, where $q$ is the prompt/question. |
| $\mathcal{R}$ | Reward model/verifier | Evaluates the reasoning quality of action $a_t$ at state $s_t$, providing feedback. |
| $r_t$ | Reward | $r_t := \mathcal{R}(s_t, a_t)$, reward given by verifier at time step $t$. |
| $\tau$ | Trajectory | $\tau := \big((s_0, a_0, r_0), \ldots, (s_T, a_T, r_T)\big)$, The entire reasoning process leading to an answer. |
| $\pi$ | Policy model/reasoner | $a_t \sim \pi(a_t\|s_t)$: The reasoning strategy that maps a reasoning state to the next reasoning step. |
| $\mathcal{V}$ | Value Model | Estimates the expected future reasoning quality from state $s_t$. |
| $\mathcal{F}$ | Refiner | $a'_t = \mathcal{F}(s_t, a_t, r_t)$: Modifies or refines the action based on feedback from the verifier. |

Table 1: An overview of symbols and terminologies for convenience.

or a single token (Schulman et al., 2017; Ouyang et al., 2022).[2] The term *Thought* generally refers to the sequence of reasoning steps (i.e., reasoning trajectory) that occur from the question (excluding the question itself) to the final answer (excluding the final answer).

**Reasoning as MDP**  An MDP is a general framework for modeling environments where an agent makes sequential decisions by observing states and receiving rewards for its actions. The state-action-reward trajectories in an MDP can be formally expressed as: $\tau = \big((s_0, a_0, r_0), \ldots, (s_T, a_T, r_T)\big)$, where $T$ is the trajectory length. Naturally, LLM reasoning can be framed as an MDP, as each reasoning step builds upon previous ones to arrive at a final answer ($s_T$) from a question ($s_0$). However, a key distinction lies in how the state transition function $P(s_{t+1}|s_t, a_t)$ is defined. In traditional MDPs, state transitions are driven by the environment (unknown to the agent). In LLM reasoning, this depends on the system architecture: in standalone LLMs, the model itself generates the next state, whereas in agentic systems, state transitions can be influenced by external tools within the environment.

In RL-based approaches, the goal is to maximize the reasoning quality measured by the cumulative reward:

$$\max \mathbb{E}_{\tau \sim P(\tau|s_0, \pi)} \left[ \sum_{t=1}^{T} r_t \right], \tag{1}$$

where $\pi$ is the reasoning policy and $r_t = \mathcal{R}(s_t, a_t)$ is the reward given by the reward function $\mathcal{R}$ at time step $t$. There are two primary approaches to optimize Equation 1. The first is via **training**, which involves optimizing model parameters to learn the optimal policy $\pi$ through methods like preference learning (e.g., DPO (Rafailov et al., 2023)) or reinforcement learning (e.g., PPO (Schulman et al., 2017)). The second is **inference-scaling**, which optimizes Equation 1 without altering model parameters. Instead, it employs a form of "search" with a frozen model, often guided by a reward model (Zhang et al., 2025b). We summarize key terminologies in Table 1.

## 2.2  Key Components of LLM Reasoning Systems

An LLM-based reasoning system may contain three key components depending on the reasoning regime and system architecture: **(a) A Reasoner** that generates the reasoning steps, serving as the policy model; **(b) Verifiers** that evaluate the correctness of the final outcome and/or reasoning steps, serving as reward functions; and **(c) A Refiner** that improves reasoning trajectories by refining responses based on the feedback from the verifier. Figure 4 shows a depiction of these components. While these components play complementary and important roles in a reasoning system, they can be implemented by the same LLM, e.g., self-refinement (Saunders et al., 2022; Madaan et al., 2024) unifies them.

**Reasoner**  The reasoner generates reasoning steps based on the current state of the reasoning process. It takes as input the previous states and outputs the next response or action. As the core component of a reasoning system, it determines how reasoning progresses and influences the final outcome.

---

[2] Although RLHF (Reinforcement Learning from Human Feedback) methods (Ouyang et al., 2022) receive rewards based on the final answer (outcome level), the underlying RL algorithms operate as multi-step RL at the token level. This differs from approaches like DeepSeek-R1 (DeepSeek-AI et al., 2025), which employs one-step RL for training.

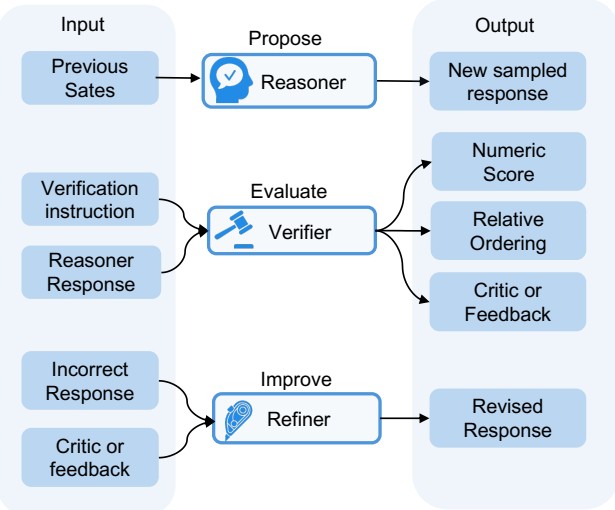

Figure 4: Three key components of a reasoning system. The *Reasoner* proposes new responses (usually accompanied with rationales) for a query. The *Verifier* takes as input a verification instruction (e.g., what aspects to evaluate) and the response(s) from the reasoner, then outputs a judgment on the response(s) (often in the form of a numeric score or relative order, and typically accompanied by a natural language critique or rationale for its judgment). The *Refiner*, unlike the first two, takes as input an incorrect response and optionally the critique (as provided by the verifier) and outputs a revised response.

**Verifier** The verifier assesses the quality of the final answer or intermediate reasoning steps and provides feedback to the reasoner. Verifiers can be outcome-level, where only the outcome is evaluated, or process-level, where intermediate reasoning steps are also evaluated. The type of feedback can range from a scalar reward (e.g., correct/wrong answer on a math problem or pass/fail for code test case) to natural language explanations. When ground-truth is available (e.g., during training), the verifier can be implemented using rule-based functions (e.g., string matching) or by training a reward model or using an LLM-judge model.

**Refiner** Given a feedback from the verifier, as well as a response from the reasoner, a refiner tries to improve and polish the original reasoning trajectory containing flaws. Refiners can play two important roles in reasoning. First, it can serve as a general approach to improve the performance during inference. More importantly, by providing explicit analysis, a refiner can also conduct implicit search, i.e., pointing out the obstacles in current trajectory, and offer a new perspective to compress the search space. Yet, recent studies (Qu et al., 2024a) show that is not at least easier than learning reasoning.

## 2.3 System Architectures

Building on the three key components introduced above, in this section, we describe how these elements are organized within different system architectures to achieve effective reasoning. While the three components serve as the foundation, their integration and interaction vary across architectural paradigms. In this survey, we structure reasoning systems into three main types: *standalone LLM*, *single-agent system*, and *multi-agent system*. Figure 5 shows their comparison with visualizations.

### 2.3.1 Standalone LLM Systems

A standalone LLM system comprises a single LLM which can play the role of one or more components (we refer this as unified components) in the reasoning system. It processes an input prompt and generates final outputs, which often include rationales or reasoning steps. As an LLM, it has the capability to produce diverse rationales through sampling—a key property utilized by many advanced reasoning techniques. Importantly, a standalone LLM operates independently, without interacting with external environments or collaborating with other LLMs. Its decision-making is based solely on simple input-output mappings or through iterative

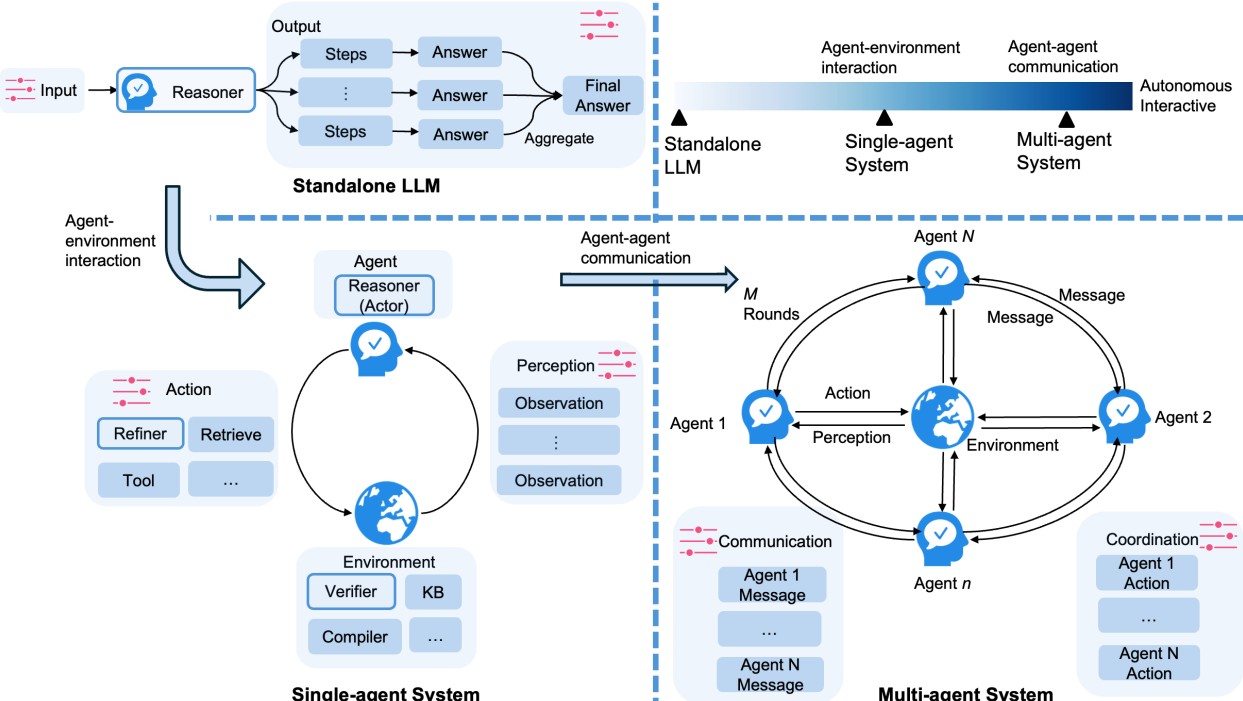

Figure 5: Three architecture types used for designing a reasoning system in the context of LLMs. ![icon] highlights perspectives that the literature emphasizes for customization.

sampling from the same model, where the prompt incorporates prior reasoning steps (a method known as self-contained reasoning). This self-contained nature allows the LLM to function autonomously while maintaining coherence in its reasoning processes.

### 2.3.2 From Standalone LLM to Language Agents

While the concept of an agent has been a long-standing idea in AI (Russell & Norvig, 2010), the notion of language agents has gained prominence alongside recent advancements in LLMs.[3] The key distinction between an agent and a standalone LLM lies in two advanced capabilities: *interactiveness* (Weng, 2023; Yao & Narasimhan, 2023) and *autonomy* (Xi et al., 2023; Wang et al., 2024d). *Interactiveness* refers to an agent's ability to engage with the external world, including environments or other agents. This capability is crucial because LLMs, while powerful, often have limited knowledge and reasoning abilities confined to their internal memory. By enabling interaction with the external world, an LLM can augment its internal knowledge with external information, significantly expanding its understanding and grounding its outputs in real-world observations. *Autonomy*, on the other hand, refers to an agent's ability not only to follow human instructions but also to independently initiate and execute actions. This capability often involves *planning* but can extend to more complex behaviors. For instance, a fully autonomous agent should be capable of detecting novel situations, proactively taking initiative, and determining effective interaction strategies without explicit human guidance. These advanced capabilities distinguish LLM-based agents from standalone LLMs, enabling them to operate more dynamically and adaptively in real-world scenarios.

To delineate the boundary between the agent and its environment, we employ the concept of *controllability* (Sumers et al., 2024). Specifically, the environment is defined as an external module that the agent cannot modify. For example, a knowledge base containing resources like Wikipedia or a compiler is considered part of the environment because the agent cannot alter it. Similarly, another LLM acting as a judge or verifier is also treated as part of the environment, as its outputs operate independently of the agent. In contrast,

---

[3]In this survey, the terms agent and LLM-based agent are used interchangeably unless stated otherwise.

components like working memory or prompts that the agent can directly modify are not classified as part of the environment.

In this work, we adopt the perspective of Kapoor et al. (2024), which conceptualizes agenticness as a *spectrum*. The more interactiveness and autonomy an LLM exhibits, the more agentic it is considered to be. In the upper right of Figure 5, we illustrate this spectrum visually. Within this spectrum, we define a system with *agent-environment interaction* as a *single-agent system* and a system that additionally incorporates *agent-agent communication* as a *multi-agent system*.

### 2.3.3 Single-agent Systems

Given the definitions above, the interaction between the agent and its environment is a central aspect of single-agent systems. These interactions can vary widely in complexity and design. In Figure 5, we illustrate a single-agent system in the bottom left. The focus here is on designing the agent's actions—such as tool use, retrieval, or answer refinement—and obtaining useful perceptions from the environment, which may include feedback from an external verifier or compiler, or data from a knowledge base (KB). This architecture enhances the LLM's capabilities by enabling it to dynamically engage with and adapt to external contexts.

While a fully autonomous agent should ideally learn to interact with the environment automatically, the literature identifies several predefined interaction patterns (also referred to as workflows (Schluntz & Zhang, 2024)) that have proven effective. We elaborate on these patterns below and, in Sections 3.2 and 5.2, explore specific techniques that leverage them to improve agent performance.

• **Generator-evaluator pattern.** This pattern divides the reasoning capability into two distinct components: a generator and an evaluator (e.g., a verifier or other evaluators like compilers). It represents a natural extension of RL-style optimization and has gained popularity since the introduction of RLHF (Ouyang et al., 2022). In this setup, the evaluator functions as the environment, providing feedback on the quality of the agent's actions. Such feedback is particularly valuable for guiding the search for effective actions and improving decision-making. Recent studies have demonstrated that verifiers can significantly enhance the performance and generalization capabilities of agents (Zhang et al., 2024i; Sun et al., 2024c). However, this pattern is not without its challenges. It can suffer from unreliable components and error propagation. For instance, Kim et al. (2024d) points out that verifiers are vulnerable to reward hacking, where the reasoner exploits loopholes in the verifier to achieve higher reward scores, ultimately degrading the overall performance of the agentic system.

• **Generator-critic-refiner pattern** This pattern divides reasoning capabilities into three components: a reasoner, a critic, and a refiner. The critic acts as the environment, providing feedback—typically in the form of guidance on how to correct errors in the generated actions. The refiner then takes the flawed actions and the critic's feedback as input, producing revised and improved actions. This pattern enables the agentic system to benefit from iterative feedback, making it particularly effective for complex tasks where the initial outputs of the reasoner are suboptimal. However, it may also lead to a phenomenon known as 'over-refinement' (Chen et al., 2024b), where the agent iterates excessively, leading to diminishing returns or even degraded performance rather than improvement. Careful design and balancing of the refinement process are essential to mitigate this risk and ensure the pattern's effectiveness.

### 2.3.4 Multi-agent Systems

In addition to the agent-environment loop in single-agent systems, multi-agent systems introduce an additional agent-agent loop, where multiple agents interact and influence one another. In this framework, agents assume different roles, exchange *messages*, and collaboratively coordinate their actions while operating within a shared environment.[4] Figure 5 shows an example multi-agent system. It involves $N$ agents (often playing distinct roles) and $M$ rounds of communication through message exchanges. The focus is on designing effective communication protocols (e.g., debates) and coordinating the agents' actions to determine a final decision or action within the environment (e.g., employing an additional judge to adjudicate final actions). The following *communication patterns* have emerged as effective predefined strategies:

---

[4]We use *message* to denote agent-agent communication and *action* to denote agent-environment interaction.

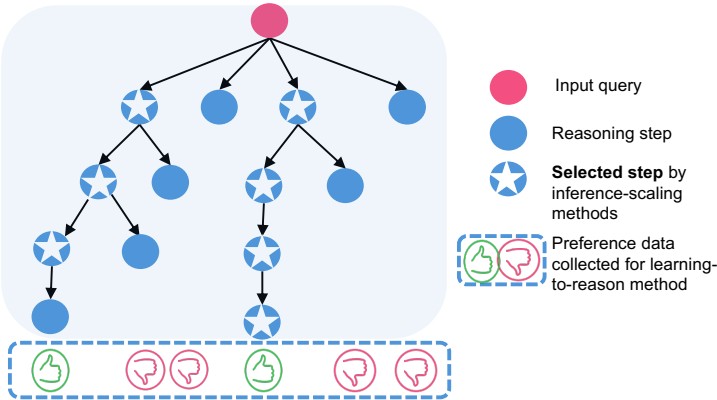

Figure 6: Inference-time and training-time regimes of a reasoning system. We use tree search as an example to illustrate the inference scaling and trajectories collection. Given a query, *inference scaling* relies on extensive inference computation to improve the reasoner's distribution. Specifically, it generates multiple candidate reasoning steps at each layer and selects the best solution to proceed (e.g., by using an external verifier or ensembling). In contrast, *learning to reason* focuses on collecting trajectories and training from the collected data with minimal inference-time computation. It takes all trajectories in the process (identical to those used in inference-scaling, allowing us to reuse the same tree) and labels them with preferences. The preference data can then be used to train the reasoner.

• **Debate pattern.** In this pattern, two or more agents engage in a debate with each other. The term debate can vary in implementation. For example, in (Wang et al., 2024h), it involves agents addressing the problem independently and incorporating other agents' responses as additional advice. In (Liang et al., 2023b), it means agents approach the problem from *opposing perspectives.* After the debate, a consensus is reached through mechanisms such as an additional judge, weighted voting, or a fixed number of iterations, ultimately determining the collective action to be taken in the environment.

• **Reconcile pattern.** This pattern facilitates collaborative round-table discussions among agents, enabling them to reach a consensus through mechanisms such as voting or confidence levels. For instance, ReConcile (Chen et al., 2023c) introduce a round-table discussion framework where agents make decisions using a weighted voting system. In this process, each agent assigns a confidence level to its proposed answers, and these confidence levels are used as weights to cast votes, ultimately determining the final decision.

## 2.4 Reasoning Regimes

Orthogonal to the components and architectures discussed above, reasoning systems can operate under distinct computational regimes. Systems employing inference-time computation can refine their outputs through iterative reflection and revision or search for improved solutions by repeatedly sampling the underlying model. However, such systems must balance cost (e.g., computational resources, latency) and effectiveness (e.g., accuracy, reliability) in achieving correct solutions. The learning-to-reason paradigm addresses this tradeoff by shifting computational burdens from inference to training, learning policies from simulated reasoning processes. While both regimes enhance effectiveness by redistributing computational effort across training and inference, they lack the capacity to dynamically adapt resource allocation or method selection to individual problems—a limitation highlighted in recent work (Sprague et al., 2024a; Kapoor et al., 2024; Chen et al., 2024d). To bridge this gap, emerging approaches within the learning-to-reason framework focus on optimizing the reasoning process itself, jointly minimizing cost and maximizing effectiveness. This involves dynamically allocating computational resources, searching for contextually optimal methods, and training models to synergize with adaptive inference-time strategies. Figure 6 contrasts these regimes, and we elaborate on each in the sections below.

| Perspective | Method | Characteristic | Representative Work |
|---|---|---|---|
| Constructing Prompts | Instruction engineering | Modify instruction by human-design template | Paranjape et al. (2021); Zhou et al. (2023b) |
| | Demonstration engineering | Drawing analogy from relevant experience | Wei et al. (2022b); Luo et al. (2024d) |
| | Prompt optimization | Search for optimized prompt (e.g., bootstrap) | Xu et al. (2022); Pryzant et al. (2023) |
| Optimizing Output | Generating subtasks | Decompose the original task into manageable subtasks | Dua et al. (2022); Zhou et al. (2023a) |
| | Exploration and search | Branch and explore multiple paths to optimize reasoning trajectories | Yao et al. (2023a); Besta et al. (2024) |

Table 2: Summary of inference scaling with standalone LLM.

### 2.4.1 Inference Scaling

Inference scaling techniques enhance reasoning capabilities during test time by increasing the amount of computation performed before generating an answer. These methods can be broadly categorized into three key strategies: (a) Prompt engineering and optimization, which focuses on constructing effective reasoning-provoking prompts through template-based methods, human curation, and automated optimization. (b) Search and planning methods, which include task decomposition, plan generation and verification, and exploration-based approaches. They enable structured multi-step reasoning, often involving backtracking within trees or graphs, to systematically explore potential solutions and verify their validity. (c) System-level enhancements, which incorporates external tools, knowledge sources, and verification mechanisms to augment the model's reasoning capabilities. For standalone LLMs, inference scaling primarily revolves around prompt construction and search strategies. In multi-agent settings, it further extends to include agent-agent communication and coordinated action strategies, enabling collaborative problem-solving. While these techniques have demonstrated significant effectiveness in improving reasoning performance without requiring updates to model parameters, they often come with increased computational costs during inference.

### 2.4.2 Learning to Reason

This regime shifts the focus to training models to reason effectively before deployment, often referred to as *training-time methods*. The core idea is to simulate inference, generating trajectories that capture potential reasoning paths. These trajectories are then used to train the reasoner with online or offline learning methods. The methods include supervised and/or reinforcement learning. While learning-to-reason typically minimizes computational costs during inference, it incurs higher costs during simulation and training. In Section 5, we provide a detailed discussion of methods within this regime across different architectures.

Recently, this paradigm has evolved to incorporate knowledge of both training and testing methods, enabling adaptive strategies. For instance, it now allows for the training of reasoners optimized for known inference techniques (Balashankar et al., 2024), or dynamically distributes computational costs between training and testing, offering a more flexible and efficient framework (Damani et al., 2025; Yue et al., 2025).

## 3 Improving Reasoning with Inference Scaling

Compared to small-scale models, pretrained large-scale language models (LLMs) have demonstrated emergent capabilities (Wei et al., 2022a), such as in-context learning (Dong et al., 2024) and role-playing (Shanahan et al., 2023a), which manifest without additional fine-tuning (i.e., without any gradient updates). Arguably, many of these abilities become apparent only after reaching a certain scale in model size. While scaling model parameters has been shown to improve reasoning performance across various tasks, the returns have diminished due to the high cost of training increasingly larger models. As a result, **inference scaling** has emerged as an appealing and orthogonal paradigm to unlock reasoning abilities in LLMs by providing additional test-time compute, allowing them to "think" before producing a final answer. It has been demonstrated that optimal scaling of test-time compute can be more effective than scaling model parameters (Snell et al., 2024), as it offers better generalization through enhanced flexibility in prompt and workflow design. Such deliberate thinking can be enabled either through training (DeepSeek-AI et al., 2025) or by explicit programming at inference time (OpenAI et al., 2024). In this section, we focus on the latter and defer training-time methods to Section 5. We begin with inference scaling methods for standalone LLMs and subsequently extend the discussion to single and multi-agent compound systems.

## 3.1 Inference Scaling With Standalone LLM

In this section, we examine the core components and techniques that have made inference-time reasoning methods effective. Many of these methods draw inspiration from research on human cognitive processes on planning, problem solving, and decision-making (Newell et al., 1959; 1972; Stanovich & West, 2000).

### 3.1.1 Constructing Reasoning Provoking Prompts

Although large-scale pre-training endows LLMs with patterns that support reasoning, these capabilities often remain latent under generic prompts. Liu et al. (2025c) demonstrate that deep-reasoning behaviors—such as reflection and self-verification, which signal profound analytical thought—can be amplified simply by increasing the sampling budget. This highlights the importance of designing prompts that deliberately provoke reasoning, thereby surfacing and leveraging the latent human priors within LLMs.

**Instruction engineering** Enabling LLMs to reason effectively depends heavily on the quality of the instructions provided (Sclar et al., 2024; Zhuo et al., 2024; Long et al., 2024a). Recognizing this, numerous prompt engineering studies aim to improve LLM reasoning by enhancing instructions. Extensive efforts in this direction primarily focus on template-based and human-curated instructions (Paranjape et al., 2021; Sanh et al., 2022; Mishra et al., 2022; Si et al., 2023; Long et al., 2024b). With LLMs becoming increasingly adept at following human instructions and generating human-like text, focus has shifted toward leveraging the models themselves to craft and refine high-quality instructions. A notable example of this shift is the Automatic Prompt Engineer (APE) introduced by Zhou et al. (2023b), which uses LLMs to generate high-quality instructions, achieving performance comparable to or surpassing that of human annotators on 31 reasoning tasks. Furthermore, other studies have proposed methods to modify instructions for improved reasoning. For instance, Deng et al. (2023a) and Mekala et al. (2024) present Rephrase-and-Response and EchoPrompt, respectively, two simple yet effective strategies where LLMs are instructed to rephrase queries before answering, significantly enhancing LLM performance on reasoning tasks. Similarly, Tian et al. (2023) introduce R3 prompting, which instructs LLMs to first extract key sentences from noisy contexts, then rephrase the instruction to explicitly include extracted sentences.

**Demonstration engineering** Humans can address new problems by drawing *analogy* from relevant past experience (Holyoak, 2012). Inspired by this, Yasunaga et al. (2024) propose analogical prompting to guide LLMs to self-generate exemplars or knowledge relevant to the given problem as few-shot demonstrations for reasoning, outperforming hand-crafted or retrieved examples. For example, LLMs are prompted to generate a problem on calculating a third-order determinant before solving the given fourth-order determinant. Similarly, Chen et al. (2023d); Yang et al. (2023a); Luo et al. (2024a) highlight the effectiveness of self-generated relevant exemplars. Qin et al. (2025) further systematically assess the capability of LLMs to perform *analogical reasoning* and find that performance is not primarily determined by whether the exemplars are topically relevant to the task. Instead, they show that even exemplars from unrelated domains, such as self-generated biological exemplars, can lead to improved performance, as long as they are accurate and structurally aligned with the reasoning steps required by the target task. This highlights that the quality of the exemplar (its correctness, clarity, and structural usefulness for reasoning) can be the key limiting factor, rather than the relevancy regarding to the topic domain.

Conventionally, a fixed set of few-shot demonstrations is applied to all queries, which can be suboptimal, especially when queries vary significantly. An alternative approach is to retrieve demonstrations tailored to the current query. Research has shown that retrieval-based demonstration selection significantly improves task performance. The main goals for selecting demonstrations are *similarity* (Rubin et al., 2022; Agrawal et al., 2023; Li et al., 2023e; Ye et al., 2023a) and *diversity* (Levy et al., 2023; He et al., 2023; Kim et al., 2024a). Various retrieval strategies have been proposed for selecting $k$ demonstrations, including top-$k$ similarity-based retrieval (Liu et al., 2022; Li et al., 2023e), clustering-based retrieval (Luo et al., 2023c; Wang et al., 2024i), and iterative retrieval (Khattab et al., 2022; Levy et al., 2023; Wang et al., 2024e). These methods enable adaptive and effective demonstration selection, enhancing the model's reasoning and generalization across diverse queries.

In addition, many-shot in-context learning has emerged as a complementary line of work, where hundreds or even thousands of demonstrations are provided to significantly enhance the performance of LLMs, especially on complex reasoning tasks (Li et al., 2023c; Agarwal et al., 2024; Zou et al., 2024; Gu et al., 2025). Many-shot prompting can be seen as an extreme form of demonstration engineering, where the focus is on scaling the quantity of demonstrations to maximize the model's capacity to learn from in-context examples. However, the effectiveness of many-shot ICL is often limited by the high cost of obtaining a large number of labeled demonstrations. To mitigate this gap, Chen et al. (2025) recently introduce MAPLE, a novel influence-based many-shot ICL framework that identifies impactful unlabeled samples, pseudo-labels them by querying LLMs, and adaptively selects them for each test query. This approach effectively enhances many-shot ICL performance with minimal labeling cost, demonstrating improved adaptability and reasoning capabilities of LLMs.

**Prompt optimization** Prompt optimization methods, aiming to systematically and strategically optimize prompts for improved performance, have been extensively explored for enhancing LLM reasoning. For instance, Xu et al. (2022) introduce Genetic Prompt Search (GPS), leveraging genetic algorithms to search for the best instruction. Similarly, Guo et al. (2024a) and Fernando et al. (2024) employ evolutionary algorithms to iteratively refine instructions, while Long et al. (2024c) introduce a minimax-game framework, inspired by Generative Adversarial Networks (Goodfellow et al., 2014) to simultaneously optimize instructions and demonstrations. Furthermore, Pryzant et al. (2023) present the concept of "text gradients" which leverage feedback from prompt executions and LLMs to update prompts, akin to Optimization by PROmpting (OPRO) (Yang et al., 2024c), which uses execution feedback. Despite these advances, the interplay between various prompt optimization algorithms remains underexplored. Recently, Wan et al. (2024a) conducted a comprehensive evaluation of representative techniques for instruction and demonstration optimization, examining their effectiveness in isolation and combination across a range of challenging tasks. Their findings indicate that intelligently reusing samples from prompt evaluations as demonstrations consistently enhances performance, that demonstration selection strategies can have a greater impact than instruction optimization techniques, and that a synergistic combination of demonstration and instruction optimization can outperform their individual contributions.

### 3.1.2 Optimizing Reasoning Output with Search and Planning

**Generating reasoning subtasks** Human problem-solving often involves planning manageable steps that lead to a successful resolution (Dostál, 2015). Likewise, improving LLM reasoning by breaking down complex problems into intermediate steps has become a successful paradigm. In this context, *subtasks* refer to the decomposed parts of a problem, *structures* are the frameworks guiding the reasoning process, and *intermediate steps* are intermediate results produced at each stage of problem-solving. Nye et al. (2021) and Wei et al. (2022b) pioneer this direction by proposing Chain-of-Thought (CoT) prompting which uses a few demonstrations with human-written intermediate steps to guide the model in solving complex problems in a similar style. Kojima et al. (2022) further simplified this approach by introducing zero-shot CoT prompting, which eliminates the need for demonstrations by instructing models to "think step by step" before answering.

Simple CoT prompting often struggles as task complexity increases, particularly when the task surpasses the complexity of the provided demonstrations. To address this, researchers have proposed methods that explicitly guide models in decomposing tasks into subtasks, thereby enhancing intermediate step reasoning. Dua et al. (2022) propose an iterative approach, where tasks are progressively broken down into simpler subtasks and solved step-by-step. Similarly, Zhou et al. (2023a); Khot et al. (2023) and Suzgun & Kalai (2024a) advocate for a "divide-and-conquer" strategy, where tasks are first divided into subtasks and then solved sequentially.

Beyond subtasks, researchers emphasize the importance of robust reasoning structures such as hierarchical and decision-making processes that capture the underlying mechanisms involved in problem-solving. Zhou et al. (2024b) introduce Self-Discover, a framework that enables models to self-identify reasoning structures for any task using a seed set of general reasoning skill modules. Building on this, Aswani et al. (2024) propose Auto-Evolve, which dynamically adapts reasoning modules to accommodate more diverse problems. In addition to designing better reasoning steps, several studies address the need to correct intermediate

| Perspective | Method | Characteristic | Representative Work |
|---|---|---|---|
| Feedback Refinement | Verifier and Reflection | Use verifiers to select, modify, or refine actions | Snell et al. (2025); Madaan et al. (2023b) |
| Action Enhancement | Retrieval and Tool | Access external knowledge and specialized resources | Li et al. (2024e); Ma et al. (2024a) |

Table 3: Summary of inference scaling with single-agent system

steps. For example, Deng et al. (2024a); Yan et al. (2024) and Wu et al. (2024b) propose methods to refine intermediate outputs. Notably, Zhang et al. (2024i) observe that smaller models ($\leq$ 13B parameters) in particular need stronger models acting as verifiers to validate and correct intermediate steps.

**Exploration and search**   Research on human problem-solving reveals that complex reasoning tasks often admit multiple valid paths to reach a correct solution (Stanovich & West, 2000). Compared to linear reasoning structures like chain-of-thought, approaches that incorporate exploration during problem-solving have shown significant improvements for complex reasoning tasks. Unlike task decomposition methods (Dua et al., 2022; Zhou et al., 2023a; Khot et al., 2023), exploration-based approaches employ dynamic search through multiple possible reasoning paths simultaneously rather than following certain decomposition patterns, enabling models to explore ambiguous solution strategies for complex problems. Exploration typically involves two key components: branching and aggregation. Due to the stochastic nature of language model decoding, branching is often implemented through independent re-sampling with non-zero temperature, generating diverse reasoning chains. Early methods, such as self-consistency (Wang et al., 2023f), introduced branching only at the beginning of the reasoning chain, conditioned on the initial query. While simple, this approach lacks local exploration of intermediate reasoning steps, has limited applicability for tasks with multiple valid answers, and produces reasoning chains with restricted diversity (Chen et al., 2024d). More recent advancements, such as Tree-of-Thoughts (Yao et al., 2023a), Graph-of-Thoughts (Besta et al., 2024), and Forest-of-Thoughts (Bi et al., 2024), enable finer-grained branching by considering both the query and a history of previous thoughts or thought-state sequences, allowing for more nuanced and flexible exploration.

The effectiveness of branched reasoning paths with thoughts or answers depends on aggregation or evaluation strategies. Recent progress is centered around two categories: ensemble-based methods and verifier-based methods. Ensemble-based methods have been widely employed due to their simplicity and self-contained nature, requiring no external knowledge or sources for validation. These approaches typically employ strategies such as majority voting across answer tokens (Wang et al., 2023f; 2024a; Li et al., 2024b) or confidence-based selection (Wang & Zhou, 2024). Verifier-based methods, in contrast, employ external verifiers or judges to score and select preferred answers among candidate solutions.

## 3.2   Inference Scaling With Single-agent System

LLMs are trained on static, finite datasets, which inherently limits their parametric knowledge. This limitation hinders their ability to reason effectively in scenarios requiring up-to-date or highly specialized knowledge. The use of an agentic system, where LLMs are augmented with external verifiers, retrieval and tool integration, has proven effective in such scenarios. Verifiers provide reasoners with a signal of the quality of their outputs (e.g., a score or natural language feedback), which may be used by reasoners to modify or improve their outputs. Retrieval augmentation improves reasoning by enabling the agent to access relevant external knowledge, thereby reducing hallucinations and ensuring more accurate, fact-based responses. Additionally, the agent can achieve higher performance by leveraging specialized external tools to handle specific intermediate reasoning steps. For instance, allowing an agent to use a calculator can minimize errors stemming from inaccuracies in numerical generation.

A pioneering approach in this domain is the ReAct framework (Yao et al., 2023b), which interleaves reasoning and acting by prompting LLMs to generate both reasoning traces and task-specific actions in an interleaved manner. This synergy allows the model to induce, track, and update action plans while interfacing with external sources (environment) to gather additional information. ReAct has demonstrated effectiveness across QA and interactive decision-making tasks. Building upon ReAct, LATS (Zhou et al., 2024a) unifies reasoning, acting, and planning within LLMs. By combining Monte Carlo Tree Search with ReAct, LATS enables structured search over a combinatorial space of reasoning and acting paths. More recently,  Liu et al.

(2024f) formalize reasoning and acting with LLMs under a Bayesian adaptive MDP and propose RAFA, a theoretically grounded framework for orchestrating the reasoning and acting of LLMs.

### 3.2.1 Refinement with Verifiers and Reflections

A natural basis for modifying agent actions is the quality of their generated outputs—if the output is incorrect, the agent should attempt to correct it. However, ground-truth references are typically unavailable to the agent at test time. In such scenarios, agents often rely on *verifiers*, which are models or systems that provide an approximate measure of correctness, to guide action modifications. A special case arises when the verifier has access to ground-truth outcomes. Oracle verifiers (First et al., 2023; Xin et al., 2024a), which leverage correct answers, have shown significant performance improvements over baselines without verifiers (Huang et al., 2024a; Brown et al., 2024). However, their applicability is limited to scenarios where ground-truth data is readily available or easily accessible, such as in games or structured environments.

In contrast, non-oracle (or imperfect) verifiers provide a more widely applicable solution. Their form varies depending on the task and knowledge source. For instance, Cobbe et al. (2021); Feng et al. (2023b); Snell et al. (2025) employ trained outcome reward models (ORMs) as verifiers to rerank responses. For more granular evaluation, Lightman et al. (2024) and Zhang et al. (2025b) train process reward models (PRMs) to serve as inference-time verifiers. By enabling the reward model to assess each reasoning step individually, PRMs generally yield greater improvements during inference compared to ORMs (Uesato et al., 2022; Tian et al., 2024).

While reward models provide actionable signals about the quality of model responses, they are *non-generative* verifiers. As a result, they are unsuitable for verification approaches that require natural language feedback. For instance, synthesizing unit tests (Chen et al., 2023b; Hassid et al., 2024; Kapoor et al., 2024; Cook et al., 2024), commonly used in code generation tasks, necessitates verifiers capable of generating natural language. Broadly, generative verifiers are referred to as either critique models or LLM-as-judge models. In both cases, LLMs are either prompted or fine-tuned specifically for critique and evaluation. These models have been employed not only for output reranking (Vu et al., 2024) but also for providing valuable natural language feedback (Shinn et al., 2024; Shridhar et al., 2024; McAleese et al., 2024). However, recent studies have found that LLM-as-judge models generally underperform reward models (RMs) in terms of verification (Zhang et al., 2024e). To address this, researchers have sought to combine the strengths of both approaches under the Generative RM framework (Zhang et al., 2024e; Mahan et al., 2024; Liu et al., 2025b), aiming to unify the advantages of generative feedback with the precision of reward-based evaluation.

Self-reflection or self-refinement approaches (Saunders et al., 2022; Madaan et al., 2024) aim to eliminate the need for additional, specialized verifier models by enabling the agent to critique and refine its own outputs. While some studies (Saunders et al., 2022; Madaan et al., 2024) have demonstrated empirical success, others highlight poor performance in the absence of robust verifiers (Stechly et al., 2023; Huang et al., 2024a; Stechly et al., 2024; Valmeekam et al., 2023; Shridhar et al., 2024). For a comprehensive review of recent advancements, see (Pan et al., 2024b).

While verification methods can be deployed across a wider range of domains, they are susceptible to false positives—incorrect solutions that nevertheless pass verification. This limitation becomes particularly relevant when scaling up inference compute, as it can lead to diminishing returns on computational investment. Interested readers can refer to (Stroebl et al., 2024) for a comprehensive analysis of these trade-offs.

### 3.2.2 Enhancement through Retrieval and Tool Utilization

During the reasoning process, agents can retrieve external knowledge to refine their internal state representations, resulting in more accurate reasoning steps. The advantages of retrieval are particularly pronounced in knowledge-intensive tasks that demand multi-hop and long-horizon reasoning, where connecting multiple pieces of information is essential to arrive at a final answer. Through retrieval, agents can access intermediate information, verify connections between data points, and integrate them into their reasoning process (Shi et al., 2024; Jiang et al., 2024b; Wang et al., 2024m). Retrieval also addresses critical flaws in LLMs, such as hallucination and factual inaccuracies. By grounding responses in retrieved facts, models are less prone to generating erroneous information and more likely to produce reliable and trustworthy outputs. For

| Perspective | Method | Characteristic | Representative Work |
|---|---|---|---|
| Designing | Decentralized | No hiearchy among agents | Chen et al. (2023c); Chang (2024) |
| Communication | Centralized | Presence of a central lead agent | Suzgun & Kalai (2024a); Pan et al. (2024a) |
| Action | Conditioned generation | Perform reasoning based on other agents' outputs | Wang et al. (2024c); Gao et al. (2024b) |
| Coordination | Dynamic adaptation | Adapt actions based on specific tasks | Fourney et al. (2024); Yuan et al. (2024c) |

Table 4: Summary of inference scaling in multi-agent systems.

instance, frameworks such as Verify-and-Edit (Zhao et al., 2023) and Chain-of-Knowledge (Li et al., 2024e) dynamically incorporate structured and unstructured knowledge sources to revise and correct intermediate reasoning steps within a reasoning chain. CRP-RAG (Xu et al., 2024b) improves multi-hop reasoning by dynamically adjusting reasoning paths and aggregating relevant knowledge. SelfRewardRAG (Hammane et al., 2024) enhances medical reasoning by combining RAG with self-evaluation, dynamically retrieving and synthesizing up-to-date medical information to ensure accurate response generation. By leveraging real-time data, such as clinical records from PubMed, it ensures responses are both current and precise. Another example is Think-on-Graph (Sun et al., 2023), a retrieval framework that integrates knowledge graphs (KGs) and text retrieval to deepen and refine reasoning in LLMs. GRATR (Zhu et al., 2024b) applies RAG techniques to enhance reasoning in multiplayer games with incomplete information.

In addition to search and retrieval, agents can utilize other specialized tools to overcome their inherent limitations and significantly enhance reasoning performance. By integrating tools such as calculators, compilers, calendars, or specialized APIs, agents can access domain-specific resources, enabling them to operate more effectively in targeted applications (Yu et al., 2023b; Lu et al., 2024a; Li et al., 2025a). For instance, SCIAGENT (Ma et al., 2024b) leverages domain-specific tools like SymPy and WolframAlpha to enhance the reasoning capabilities of LLMs in scientific domains. Similarly, FinAgent (Zhang et al., 2024g) combines textual, numerical, and visual tools to improve performance in financial trading tasks.

Moreover, external tools provide precise computational capabilities, allowing LLMs to transcend their limitations and perform complex numerical tasks with higher accuracy (Chen et al., 2023e; Li et al., 2023a). For example, MATHSENSEI (Das et al., 2024) employs tools such as Python, WolframAlpha, and Bing Search to tackle mathematical reasoning tasks across disciplines like algebra and calculus. TART (Lu et al., 2024b) integrates LLMs with tools for precise table-based reasoning tasks, such as table question answering and fact verification.

Moreover, Anthropic introduced an open standard of Model Context Protocol (MCP) to seamlessly connect AI assistants with real-world data sources such as content repositories, business tools, and development environments. It provides a universal, scalable way for developers to create secure, two-way connections between AI tools and diverse data systems. While MCP holds significant promise, its adoption also introduces several challenges that must be addressed to support sustainable growth and responsible development. Hou et al. (2025) discussed some key issues, such as the absence of centralized security oversight, gaps in authentication and authorization, and difficulties in maintaining consistency across multi-step, cross-system workflows.

## 3.3 Inference Scaling With Multi-agent Systems

By strategically designing communication patterns and coordinating actions, multi-agent systems can achieve more sophisticated reasoning by harnessing the specialized capabilities of multiple agents (Guo et al., 2024b). Effective communication design involves establishing structured message exchanges and interaction patterns among agents, while action coordination focuses on reconciling diverse outputs and achieving consensus to determine the final action in the environment.

### 3.3.1 Designing Communication Patterns

A common communication pattern in multi-agent frameworks involves engaging multiple agents in debates or discussions (Liang et al., 2023b). For instance, the RECONCILE framework (Chen et al., 2023c) requires each agent to generate an answer accompanied by an explanation and a confidence score. The agents then participate in multi-round discussions to refine their responses, and a confidence-weighted voting mechanism

aggregates the answers into a consensus. Similarly, SocraSynth (Chang, 2024) employs opposing LLM agents moderated by predefined contentiousness levels to explore diverse perspectives. Additionally, GroupDebate (Liu et al., 2024e) organizes agents into groups that conduct internal debates before sharing their results, reducing token costs while maintaining robust logical reasoning capabilities.

Besides decentralized communication, prior works also consider sending messages to a central node for decision making. For example, Suzgun & Kalai (2024b) employers a language model as a multi-faceted conductor that is good at handling and integrating various queries. Moreover, AgentCood (Pan et al., 2024a) assigns an LLM the role of a central planner for coordination strategy generation and agent assignment. Compared with decentralized communication, it can lead to more efficient resource allocation but increase the system vulnerability to potential failure of the central node.

### 3.3.2 Coordinating Action

Effective action coordination among multiple agents is important for achieving the shared goals, especially given a dynamic and complex environment. Prior works explore various strategies which can enable agents to synergise agents' actions and optimize overall system reasoning and problem-solving performance. This approach leverages the strengths of different LLMs to overcome the limitations of individual models.

One straightforward coordination strategy is chaining agents in a row, where agents can perform reasoning based on other agents' outputs. For example, Mixture-of-Agents (MoA) (Wang et al., 2024c) capitalizes on the cooperative nature of LLMs, allowing models to generate higher-quality responses by integrating and synthesizing contributions from multiple agents, achieving state-of-the-art performance. Similarly, Meta-Reasoning Prompting (MRP) (Gao et al., 2024b) assigns each agent to dynamically select the most effective reasoning method from a reasoning pool for a specific task, enabling the integration of diverse strategies to efficiently address multiple tasks. In addition, CoMM (Chen et al., 2024c) makes agents respond to discussions based on different role-playings.

Moreover, coordination action can incorporate dynamic adaptation to task requirements. For example, Magentic-One (Fourney et al., 2024) introduces a lead agent as Orchestrator to conduct dynamic planning based on varied tasks. Gabriel et al. (2024) proposes a framework that deals with multi-hop queries, produces and executes task graphs, chooses suitable tools, and dynamically adapts to real-time changes. Additionally, EVOAGENT (Yuan et al., 2024c) dynamically generates various agents suitable for the given task and select those with high-quality outputs for result generation.

## 4 Learning Algorithms

Before delving into methodologies for training reasoning models, we first describe the foundational learning algorithms used to train the reasoner's policy and verifiers. These algorithms are defined by their precise loss functions. Note that learning algorithms are independent of the data curation process, which will be discussed in detail in Section 5. We begin by presenting commonly used learning algorithms for training reasoning models in Section 4.1, followed by a discussion on training verifiers in Section 4.2.

### 4.1 Learning of Reasoner

This section is organized into three key parts: (1) imitation learning through supervised fine-tuning, (2) reinforcement learning, and (3) preference learning.

### 4.1.1 Imitation Learning - Supervised Fine-tuning

Supervised fine-tuning (SFT) maximizes the log probabilities of the next token $y_i$ given the input prompt $x$ and previously generated tokens $y_{<i}$. Training the policy model $\pi_\theta$ generally includes the steps to minimize the following loss function:

$$L_{\text{SFT}}(\theta) = \mathbb{E}_{x,y\sim\mathcal{D}}\left[\sum_{i}^{T} -\frac{1}{T}\log(\pi_\theta(y_i|y_{<i},x))\right], \tag{2}$$

where $\mathcal{D}$ is the SFT dataset that comprises inputs $x$ and ground truth labels $y$. The ground truth labels can be either human-written or AI-generated reasoning process and answer response. The loss is equivalent to the next token prediction objective where the prompt input tokens are masked out and do not contribute to the loss. SFT is the often the default first (or only) step to train a base LLM to produce reasoning chains in zero-shot settings. SFT has also popularly used as an effective way to train smaller LLMs to imitate outputs generated by larger, more powerful LLMs, in a process known as knowledge distillation (Xu et al., 2024c).

### 4.1.2 Reinforcement Learning for Reasoning

Stiennon et al. (2020) and Ouyang et al. (2022) pioneered the application of reinforcement learning (RL), particularly proximal policy optimization (PPO) (Schulman et al., 2017), to improve not only reasoning capabilities but also the helpfulness and harmlessness of LLMs. Their work catalyzed a wave of innovations in preference learning and RL-based optimization techniques, as evidenced by subsequent studies (Rafailov et al., 2023; Ahmadian et al., 2024; OpenAI et al., 2024; DeepSeek-AI et al., 2025; Ramesh et al., 2024).

**Markov decision process.** Most reinforcement learning (RL) approaches model text generation as a Markov Decision Process (MDP). In this framework, the process is defined by the following components:

- A set of states $\mathcal{S}$,

- A set of actions $\mathcal{A}$,

- A state-action transition distribution $P(s_{t+1}|s_t, a_t)$ controlled by the environment,

- A reward function $R(s_t, a_t) \in \mathbb{R}$ that provides a scalar reward, and

- A policy $\pi(a_t|s_t)$, which determines the actions to take based on the current state.

At each time step $t$, for a given state $s_t \in \mathcal{S}$, the agent selects an action $a_t$ and transitions to a new state $s_{t+1}$, receiving a reward $R(s_t, a_t)$ from the environment. The set of available actions at state $s_t$ may be restricted to a subset of $\mathcal{A}$, denoted $\mathcal{A}_{s_t}$ (i.e., $a_t \in \mathcal{A}_{s_t}$). In the context of autoregressive language modeling with LLMs, generally the next token depends on all the previous tokens. As such, in order to apply RL training for LLMs, one needs to define the states and actions of the problem such that they both satisfy the temporal dependency constraint of the language modeling task as well as the Markov property. One common approach is to define that the current state $s_t$ fully encapsulates all relevant information about the environment, in other words *all previous tokens*. This means the next state $s_{t+1}$ depends solely on the current state $s_t \in \mathcal{S}$ and the chosen action $a_t \in \mathcal{A}_{s_t}$. In this way, the current state no longer needs to retrieve information from the previous states to decide the next action. As such, the state transition is agnostic to the history or previous states and actions. Within this MDP framework, the goal of RL is to learn a policy model that selects optimal actions by maximizing the expected cumulative rewards (Eq. 1).

- **Action := token**: Actions are defined at the token level, making the action space $\mathcal{A}_{s_t}$ is finite and equal in size to the vocabulary. The state $s_t$ consists of all preceding tokens, including the input prompt and previously generated output tokens. The next state $s_{t+1}$ is defined as the concatenation of the current state $s_t$ and the action taken $a_t$, i.e., $s_{t+1} := [s_t; a_t]$. This category of methods defines rewards and related measures, such as values and advantages, at the token level. Works adopting this approach include most standard RLHF methods (Ouyang et al., 2022; Zheng et al., 2023b; Lee et al., 2023) as well as more recent fine-grained process-rewarding approaches (Yuan et al., 2024b; Cui et al., 2025).

- **Action := token chunk (step)**: In this category of methods, actions are defined at the level of token chunks that semantically represent a reasoning step, separated by a special delimiter. As a result, the action space is infinite. The state $s_t$ consists of the prompt and the output tokens generated in previous reasoning steps. Rewards, value scores, and advantages are computed at the step level, with all tokens within a reasoning step $a_t$ sharing the same step-level score. This approach is particularly prominent in process supervision pipelines, as exemplified by DeepSeek-Math and VinePPO (Shao et al., 2024; Kazemnejad et al., 2024).

| Type | State $s_t$ | Action $a_t$ | Action space | Example work |
|------|-------------|--------------|--------------|--------------|
| Action := token | All previous tokens (prompt and current response tokens) | one token | finite, vocabulary size | (Ouyang et al., 2022; Zheng et al., 2023b; Lee et al., 2023) |
| Action := step | All previous tokens of prompt and previous steps | a chunk of tokens representing a "reasoning step", separated by a special delimiter | infinite | (Shao et al., 2024) (process supervision), (Kazemnejad et al., 2024) |
| Action := full response | Prompt | entire response | infinite | (Shao et al., 2024) (outcome supervision), (DeepSeek-AI et al., 2025) |

Table 5: Definitions of MDP states and actions across different training schemes.

- **Action := full response**: In this category, the entire response—comprising all output tokens—is treated as a single action. This transforms the reasoning problem into a one-step MDP with an infinite action space. This approach has been recently popularized by DeepSeek-R1 (DeepSeek-AI et al., 2025) and previously by DeepSeek-Math (outcome supervision) (Shao et al., 2024). A unique aspect of this formulation is that the full response may semantically include multiple reasoning steps, such as spontaneous backtracking and self-evaluation behaviors, as observed in DeepSeek-R1 (DeepSeek-AI et al., 2025).[5] Regardless of the number of humanly recognizable reasoning steps within the response, the entire output is still considered a single action. To assign token-level value scores, rewards, and advantages, Shao et al. (2024); DeepSeek-AI et al. (2025) compute these values based on the full response $a_t$ and then distribute them uniformly across all tokens, similar to the step-level action setting. This formulation aligns with the concept of "bandit" prediction (with infinite action space) in REINFORCE-style RL (Nguyen et al., 2017; Kreutzer et al., 2017).

**Proximal Policy Optimization (PPO).** As one of the primary variants of policy gradient methods, PPO has remained a popular and widely used RL algorithm (Schulman et al., 2017). To train the policy $\pi_\theta$, PPO utilizes two additional models: the reference model $\pi_{\theta_{\text{ref}}}$, which represents the initial state of the policy, and the value model $V$, which estimates the state value $V(s_t)$. PPO begins by sampling a state-action trajectory $\tau$ with consecutive state-action pairs $s_{t+1} \sim (s_t, a_t)$, then collects the respective intermediate or process reward (if available) and final (outcome) reward. Then, it computes the advantage $A(s_t, a_t)$ of each action $a_t$ given the current state $s_t$, which is defined as the relative strength of that specific action $a_t$ compared to the probability-weighted actions that the policy could probably have taken from $s_t$. The advantage is formulated as

$$A(s_t, a_t) := Q(s_t, a_t) - V(s_t) := Q(s_t, a_t) - \mathbb{E}_{a_t'}[Q(s_t, a_t')], \tag{3}$$

where $Q(s_t, a_t)$ represents the expected cumulative total reward that the policy is expected to obtain if it takes action $a_t$ from $s_t$ and continue to follow the current policy, while $V(s_t)$ denotes the expected total rewards obtainable from state $s_t$, known as the state value. The state value is equivalent to the expected value of $Q(s_t, a_t')$ marginalized over all probable actions the current policy $\pi_\theta$ may take from $s_t$. If $A(s_t, a_t) > 0$, the action $a_t$ is encouraged, conversely, if $A(s_t, a_t) < 0$, the action $a_t$ is discouraged. After computing the advantages, PPO optimizes the policy $\pi_\theta$ according to the following loss function.

$$L_{\text{PPO}}(\theta) = \mathbb{E}_{\tau \sim \pi_{\theta_0}, P} - \frac{1}{T}\left[\sum_{t=0}^{T} min\left(\frac{\pi_\theta(a_t|s_t)}{\pi_{\theta_o}(a_t|s_t)}A(s_t, a_t), clip\left(\frac{\pi_\theta(a_t|s_t)}{\pi_{\theta_o}(a_t|s_t)}, 1 - \epsilon, 1 + \epsilon\right)A(s_t, a_t)\right)\right], \tag{4}$$

---

[5]The O-1 model series (OpenAI et al., 2024) also exhibit such behaviors, though the training approach for O-1 remains undisclosed.

where $t \in [0, T]$ is a time step within trajectory $\tau$, $\pi_{\theta_o}$ is the fixed policy from previous episode or iteration, and $P$ is the transition distribution. The *clip* function, applied to the probability ratio $\frac{\pi_\theta(a_t|s_t)}{\pi_{\theta_o}(a_t|s_t)}$, ensures that the policy does not deviate too drastically or rapidly from its previous version. This also help prevent catastrophic failure or suboptimal local solutions. Additionally, a KL divergence term $\mathcal{D}_{\text{KL}}(\pi_\theta||\pi_{\theta_{\text{ref}}})$ is often incorporated into the loss function to constrain exploration during the later stages of training. $\pi_{\theta_{\text{ref}}}$ is often a fixed initial reference policy that we do not want our policy to deviate too much from, while $\pi_{\theta_o}$ is a snapshot of the current policy from the previous iteration which is updated regularly. Throughout the training process, both the policy $\pi_\theta$ and value model $V$ are iteratively updated.

**REINFORCE & RLOO.** REINFORCE is another popular policy gradient method (Sutton, 2018; Williams, 1992; Nguyen et al., 2017; Kreutzer et al., 2017) for RL. This method seeks to optimize the reward weighted objective of the entire response as:

$$L_{\text{REINFORCE}}(\theta) = \mathbb{E}_{x\sim\mathcal{D}, y\sim\pi_\theta(\cdot|x)}[(R(y, x) - b)\nabla_{\pi_\theta}\log\pi_\theta(y|x)] \tag{5}$$

where $R(y, x)$ represents the final reward for output $y$ given input $x$ and $b$ is a baseline term introduced to reduce variance of the gradient estimates. A widely used choice for $b$ is the moving average of all rewards observed during training (Williams, 1992; Ahmadian et al., 2024).

Recently, the REINFORCE Leave-One-Out (RLOO) method (Kool et al., 2019; Ahmadian et al., 2024) has been proposed, which replaces the traditional baseline calculation with the leave-one-out average of trajectory rewards obtained through Monte Carlo (MC) sampling, as shown in Eq. 6

$$L_{\text{RLOO}}(\theta) = \frac{1}{k}\sum_{i=1}^{k}[R(y_i, x) - \frac{1}{k-1}\sum_{j\neq i}R(y_j, x)]\nabla_{\pi_\theta}\log\pi_\theta(y_i|x) \tag{6}$$

where $k$ denotes the number of Monte Carlo samples. Unlike PPO, these algorithms do not rely on a parameterized value function (critic model) and instead depend solely on observed rewards. These methods share similarities with approaches such as Group-Relative Policy Optimization (GRPO) (Ramesh et al., 2024) and VinePPO (Kazemnejad et al., 2024), which will be discussed in detail below.

**Group-Relative Policy Optimization (GRPO).** This algorithm has gained recent popularity through DeepSeek-R1 DeepSeek-AI et al. (2025), though it was also explored in earlier studies such as (Shao et al., 2024; Yang et al., 2024b;a; Team, 2024). It employs the same clipped surrogate objective as PPO, defined in Eq. 4 (Schulman et al., 2017). However, unlike PPO, which uses a parameterized value model to estimate the advantage $A(s_t, a_t)$, this approach samples a group $G = [o_1, o_2, ..., o_g]$ of Monte-Carlo outputs for a given input $x$. It then computes the corresponding rewards $R = [r_1, r_2, ..., r_g]$, and determines the advantage of each output $o_i$ as the group-normalized reward

$$A_{\text{GRPO}}(s_{i,t}, a_{i,t}) = A_{\text{GRPO}}(o_i) = \frac{r_i - mean(R)}{std(R)}. \tag{7}$$

Then, the algorithm optimizes the policy $\pi_\theta$ by minimizing the following loss function.

$$L_{\text{GRPO}}(\theta) = -\frac{1}{|G|}\sum_{i}^{|G|}\frac{1}{T_i}\sum_{t}^{T_i} min\left\{\frac{\pi_\theta(a_{i,t}|s_{i,t})}{\pi_{\theta_o}(a_{i,t}|s_{i,t})}A_{\text{GRPO}}(s_{i,t}, a_{i,t}),\right.$$
$$\left.clip\left(\frac{\pi_\theta(a_{i,t}|s_{i,t})}{\pi_{\theta_o}(a_{i,t}|s_{i,t})}, 1 - \epsilon, 1 + \epsilon\right)A_{\text{GRPO}}(s_{i,t}, a_{i,t})\right\} \tag{8}$$

Variants of GRPO, such as DAPO (Yu et al., 2025), have also been introduced to alleviate issues with GRPO like length bias and inappropriate penalties for responses that exceed the context length.

### 4.1.3 Preference Learning

Preference learning, particularly learning from human feedback, is a widely used post-pretraining alignment stage for LLMs. Its goal is to encourage the generation of responses that align with human preferences or

desired values, such as helpfulness or harmlessness (Ouyang et al., 2022; Bai et al., 2022; Ganguli et al., 2022). The data collection process for this stage typically involves prompting an unaligned LLM to generate multiple responses for a given input. Human annotators are then presented with pairs of responses and asked to select the preferred one. The resulting preference dataset is used to train a reward model. This reward model subsequently provides online reward scores for policy trajectories during PPO training, a process commonly referred to as reinforcement learning from human feedback or RLHF (Schulman et al., 2017; Ouyang et al., 2022; Touvron et al., 2023), as well as AI feedback (Lee et al., 2023).

Preference learning has evolved beyond conventional reinforcement learning (RL)-based methodologies with the introduction of Direct Preference Optimization (DPO) (Rafailov et al., 2023) and its subsequent variants (Ethayarajh et al., 2024; Lai et al., 2024; Hong et al., 2024; Saeidi et al., 2024; Meng et al., 2024; Azar et al., 2024). DPO proposes using the policy language model itself to directly model human reward preferences from the preference dataset. This formulation eliminates the need for a separately trained reward model, instead optimizing the policy on the preference dataset with a simple binary classification loss. Formally, the policy $\pi_\theta$ is optimized using a preference dataset $\mathcal{D}$ by minimizing the loss function:

$$L_{\mathrm{DPO}}(\theta) = -\mathbb{E}_{(x,y_w,y_l)\sim\mathcal{D}} \left[ \log \sigma \left( \beta \log \frac{\pi_\theta(y_w|x)}{\pi_{\mathrm{ref}}(y_w|x)} - \beta \log \frac{\pi_\theta(y_l|x)}{\pi_{\mathrm{ref}}(y_l|x)} \right) \right], \tag{9}$$

where $y_w$ and $y_l$ represent the winning (chosen) and losing (rejected) outputs for input $x$, respectively. DPO has gained popularity due to its simplicity and stability, bypassing the engineering complexity and challenges associated with PPO-based techniques. However, DPO is not without limitations, such as implicit biases toward longer responses and performance degradation over extended training periods (Ethayarajh et al., 2024; Meng et al., 2024). Subsequent advancements, including KTO (Ethayarajh et al., 2024), iPO (Azar et al., 2024), SimPO (Meng et al., 2024), ORPO (Hong et al., 2024), Step-DPO (Lai et al., 2024), and combination methods (Saeidi et al., 2024), have addressed many of these shortcomings.

While the above learning algorithms are formulated for single turn input-to-output tasks, it is also generalizable to multi-turn conversations as well as function-calling agentic workflows. In such scenarios, the next state $s_{t+1}$ may not always be a concatenation of all previous states $s_{\leq t}$ and actions $a_{\leq t}$, but it also depends on incoming response $h_t$ from an outside environment, which can come from a follow-up user instruction or the returned result from a function call. In other words, one may define $s_{t+1} := [s_t; a_t; h_t]$.

## 4.2 Learning of Verifiers and Reward Models

Verifiers play an important role in reasoning systems, improving performance both through training time credit assignment (Ouyang et al., 2022; Ziegler et al., 2019; Stiennon et al., 2020) and inference-time scaling verification (Snell et al., 2024). Reward modeling in the reasoning settings focuses on verifying the correctness of the reasoning chain, rather than evaluating using more general criteria, like helpfulness or safety (Ouyang et al., 2022). As a result, reward model training in reasoning is typically formulated as a binary classification problem between correct and incorrect reasoning steps. Based on label granularity, reward modeling is further categorized into outcome reward modeling (Section 4.2.1) and process reward modeling (Section 4.2.2). More recently, generative models for verification (Section 4.2.3) have emerged as a popular approach that produces actionable and explainable natural language feedback alongside rewards. In this section, we cover common *training* approaches for verifiers; In Section 6.1.3, we posit that verification itself may benefit from being studied as a reasoning problem itself, highlighting both concrete methods and recent analysis of failure modes in reasoning settings.

### 4.2.1 Outcome Reward Models (ORM)

The goal of outcome reward models (ORMs) for reasoning is to provide a scalar reward for a full trajectory. Given a dataset $\mathcal{D}$ of input prompt $x$ and sampled outputs $y$ with corresponding correctness label $c \in \{0, 1\}$, the goal of outcome reward modeling is to train the outcome reward model $r_\theta$ using the loss

$$L_{\mathrm{orm}}(\theta) = \mathbb{E}_{x,y\sim\mathcal{D}} \left[ c \log \sigma(r_\theta(x,y)) + (1-c) \log\big(1 - \sigma(r_\theta(x,y))\big) \right], \tag{10}$$

where $\sigma$ is the sigmoid function. Alternatively, one can train ORMs with a pairwise formulation. Here, the correctness labels are not explicitly encoded in the loss function, but are used to categorize multiple sampled

outputs as correct or incorrect. From there, we can form pairs of outputs $\{y_w, y_l\}$, where $y_w$ reaches the correct outcome (e.g., correct answer for a math problem) and $y_l$ reaches an incorrect outcome. The reward model $r_\theta$ is then typically trained with the Bradley-Terry loss, similar to that in DPO training (Equation 9).

$$L_{\text{orm}}(\theta) = -\mathbb{E}_{x, y_w, y_l \sim D}\Big[\log\big(\sigma\big(r_\theta(x, y_w) - r_\theta(x, y_l)\big)\big)\Big], \tag{11}$$

Many other pairwise loss functions can be employed, such as hinge loss or other margin-based losses, focal loss, or variations of the Bradley-Terry loss. However, recent work (Liu et al., 2024a) has categorized the impact of loss functions, finding that the typical Bradley-Terry loss yields the best-performing ORM.

### 4.2.2 Process Reward Models (PRM)

While outcome reward models are relatively simple to train, outcome-driven verification may encourage incorrect reasoning chains that lead to the correct outcome. As such, recent work has sought to train process reward models (PRMs) to assess correctness for each step in the solution. This requires more fine-grained labels than ORM training. Specifically, assume that for an output $y = (a_1, \ldots, a_T)$, we obtain process-level supervision of the form $c_1, \ldots, c_T$, where $c_t$ is a binary indicator of step $a_t$ correctness. Then, the step-wise cross-entropy loss below is applied.

$$L_{prm}(\theta) = \mathbb{E}_{x, y \sim \mathcal{D}}\left[-\frac{1}{T}\sum_{t=1}^{T}\big(c_t \log \sigma(r_\theta(x, y_{\leq t})) + (1 - c_t) \log \sigma(1 - \sigma(r_\theta(x, y_{\leq t})))\big)\right] \tag{12}$$

Above, $y_{\leq t}$ denotes the output prefix up to and including step $t$. In practice, collecting step-level annotations $c_t$ can be extremely expensive. As a result, recent work has used variants of Monte Carlo Tree Search to automatically obtain said annotations. Specifically, the annotation for a reasoning step is obtained by *rolling out* the response until completion from the intermediate step, then using the outcome accuracy as a proxy for correctness (Wang et al., 2024g; Jiao et al., 2024a; Wang et al., 2024k; Dou et al., 2024a; Luo et al., 2024b; Setlur et al., 2024b). As a concrete example, suppose we roll out five completions randomly from the same prefix $y_{\leq t}$, with three rollouts arriving at the correct answer. Then, the confidence that the prefix $y_{\leq t}$ is correct can be approximated as 0.6. These coarse signals can then be used to train a PRM. These two general approaches to constructing PRM training data have associated pros and cons: Collecting human annotations is expensive, but does not overfit PRM training to one particular policy. MCTS-based approaches yield annotations relatively quickly, but do not generalize beyond the policy from which samples are collected (Zheng et al., 2024; Setlur et al., 2024a).

### 4.2.3 Generative Verifiers

ORMs and PRMs are discriminative verifiers, and are therefore unable to generate natural language to support their scores. However, natural language reasoning for evaluations is valuable both as actionable feedback and as an explainable mechanism. As a result, generative verifiers have been proposed to assess responses and provide natural language feedback. Generative verifiers have progressed from prompting frontier LLMs to evaluation-specific finetuning, relying on many of the same learning algorithms presented in Section 4.1. As such, the focus of this section is largely on training data curation.

**Finetuned generative verifiers** Generative verifiers are broadly classified as critique models or LLM-as-judge models. Critique models typically take as input a question and model response, and produce a critique with actionable feedback in natural language. The foundation of critique model training is critique training data. To construct training data, intentionally incorrect outputs are sampled from a policy model. Then, these outputs are corrected, usually with stronger model or human annotations. Using such samples, past methods (Wang et al., 2023c; Xi et al., 2024) have employed SFT (Section 4.1.1) to train critique models to imitate critiques. Other methods (Yao et al., 2023c; McAleese et al., 2024) have used used the typical RLHF workflow (Section 4.1.3), first training a reward model to use during PPO training. More recently, outcome-based RL (e.g., GRPO, as presented in Section 4.1.2) has been used for training, relying on either hand-crafted rewards (Akyürek et al., 2023) or execution feedback for code critique (Xie et al., 2025).

LLM-as-judge models are a more general class of generative verifiers trained to evaluate model responses based on different protocols (pairwise evaluation, 1-5 rating, binary classification). These models rely on preference datasets, either annotated by a strong model or by humans. For example, to train a pairwise LLM-as-judge, one would collect a dataset of paired model responses for a given input prompt, then ask either a human or strong LLM to pick which response is better. Then, natural language explanations are distilled from stronger models, with distilled samples being categorized as correct or incorrect if the preference matches the annotation. From here, earlier LLM-as-judges (e.g., (Li et al., 2023b; Zheng et al., 2023a)) trained with SFT (Section 4.1.1), while newer approaches (Wang et al., 2024f; Hu et al., 2024) have used DPO (Section 4.1.3).

**Discriminative-generative hybrid verifiers** Because generation is a more difficult task than classification, generative verifiers have often lagged discriminative reward models in benchmark performance. Recent work (Zhang et al., 2024f; Mahan et al., 2024) has sought to unify the two under the Generative Reward Model umbrella. Here, models use similar datasets to those used to train LLM-as-judge models, but augment the SFT loss with an answer-token loss. Concretely, given a dataset $\mathcal{D}$ with samples comprised of an input $x$, model response $y$, and outcome label $c$ (e.g., "Yes"/"No" for correctness), the loss

$$L_{GenRM}(\theta) = -\mathbb{E}_{x,y,c\sim\mathcal{D}}\left[\log(\pi_\theta(c|x,y)\right] \tag{13}$$

is added to the typical language generation losses (e.g, SFT or DPO loss) that are used to train the model to produce natural language explanations. Here, $\pi_\theta$ is the generative reward model being trained.

## 5 Learning to Reason

In Section 3, we explored various methods for enhancing reasoning through inference-time computation. While these approaches have proven effective in many scenarios, they come with notable limitations, such as constrained improvements in reasoning capabilities (since model parameters remain unchanged) and the requirement for substantial computational resources during inference. With the advent of OpenAI o1 (OpenAI et al., 2024), there has been a growing emphasis on improving reasoning through training-time methods. Recently, Deepseek-R1 (DeepSeek-AI et al., 2025) demonstrated that training-time approaches can achieve reasoning improvements comparable to, or even surpassing, those of inference-scaling methods. Reflecting this trend, this section delves deeper into the role of training in advancing reasoning capabilities.

Specifically, we explore the *data recipe*, which focuses on constructing data (reasoning trajectories) tailored for reasoning tasks to facilitate training. At a high level, trajectory collection can be viewed as a form of simulation, where the generator produces reasoning steps—potentially incorporating calls and outputs from external tools—in response to either synthetic or real-world inputs. The primary challenge lies in ensuring that this simulation is both *realistic and diverse* while simultaneously *providing meaningful supervision (reward)* throughout the process. Depending on the architecture, as outlined in Section 2.3, this typically involves designing inputs (such as perception in single-agent systems or interaction in multi-agent systems) and outputs (such as actions in single-agent systems or coordination in multi-agent systems).

Furthermore, we explore the *model recipe*. Depending on the learning algorithms (Section 4), the model recipe can be *'offline'* (non-RL, e.g., SFT and offline RL, e.g. DPO), which focuses on extracting supervision (reward) from the collected trajectories and leveraging them for training. It can also be *'online'* (most of RL algorithms, e.g., GRPO and PPO), where there is no need to collect trajectories beforehand, but learning occurs directly on the questions and their rewards. Similar to Section 3, we start with standalone LLMs, detailing how each of their components is trained (Section 5.1). Building on this foundation, we expand the discussion to single-agent systems (Section 5.2) and multi-agent systems (Section 5.3).

### 5.1 Learning to Reason with Standalone LLM

This section examines how standalone LLMs can be trained for reasoning tasks. For 'offline' methods, the process typically involves collecting reasoning trajectories, that lead to both correct and incorrect outcomes, followed by further training the LLM on these trajectories. In contrast, for 'online' methods, learning occurs

| Perspective | Method | Characteristic | Representative Work |
|---|---|---|---|
| Constructing Prompts | Question Augmentation | Expand knowledge depth and breadth of seed questions | Luo et al. (2023b); Yu et al. (2024c) |
| | Graph-based Synthesis | Synthesize prompts guided by structured taxonomy | Li et al. (2024a); Tang et al. (2024) |
| Collecting Trajectories | Rejection Sampling | Filter low-quality trajectories from current policy | Dong et al. (2023) |
| | Special Reasoning Pattern | Imitate human-like reasoning behavior | Yuan et al. (2024a); Qin et al. (2024) |
| | Reasoning Distillation | Distill reasoning capability from frontier reasoning model | Huang et al. (2024d) |
| Training from Trajectories | Imitation Learning | Learn the behavior directly from the collected trajectories | Yu et al. (2024c) |
| | Preference Learning | Optimize preference between pos. and neg. trajectories | Jiao et al. (2024a) |
| | Latent Reasoning | Compress trajectory length using implicit reasoning tokens | Hao et al. (2024b) |

Table 6: Summary of learning to reason with standalone LLM.

directly based on the sampled reasoning chains and their corresponding rewards. While much of the research focus has been on sampling high-quality outputs (i.e., trajectories), methods for generating a robust and diverse set of problems, or model inputs, have also garnered attention. We begin by detailing the process of collecting trajectories, which includes constructing inputs (Section 5.1.1) and obtaining outputs (Section 5.1.2). Subsequently, we describe how the LLM can be trained using the collected trajectories (Section 5.1.3).

### 5.1.1 Constructing High-quality Prompts for Reasoning

To effectively drive knowledge distillation and model-seeking, we must curate a diverse collection of high-quality prompts that comprehensively span the target knowledge space. Relying on a narrow or homogeneous prompt set—even when sourced from a strong base model—limits exploration and undermines both distillation and reinforcement learning processes. By contrast, carefully crafted prompts expand the model's exploratory capacity, yielding richer representations and more robust downstream performance. As such, this section covers methods for collecting or synthesizing more challenging prompts.

**Question augmentation**  A straightforward approach to generating additional inputs is to directly augment existing datasets using frontier LLMs. For example, Xu et al. (2024a) propose using LLMs to "evolve" existing prompt sets, expanding their depth (e.g., more complex instructions) and breadth (e.g., rarer concepts). Yu et al. (2024c) have proposed two main approaches to augment existing questions. One is simply rewriting using frontier LLMs, and the other one is self-verification, which transforms an condition in the question into unknown variable, shows the original answer, and proposes a new question by querying the value of the unknown variable. Luo et al. (2023b) adopt a comparable strategy, employing a question generator to iteratively produce both harder and easier versions of a given question, as inspired by the instruction evolution approach of Xu et al. (2024a). The synthesized instructions are further refined using a reward model to ensure quality.

**Knowledge graph-based synthesis**  Directly augmenting prompts with LLMs can increase the size of the training set but does not inherently enhance diversity. To address this, knowledge graphs—structured taxonomies for organizing reasoning domains—have been utilized to construct input prompts with broader coverage. For instance, Li et al. (2024a) employ a frontier LLM to generate a knowledge graph directly, while Tang et al. (2024) task a frontier LLM with extracting a taxonomy from a seed dataset. These knowledge graphs are then used to progressively synthesize challenging questions, which are subsequently used to prompt larger teacher LLMs, resulting in high-quality instruction-tuning datasets with wider knowledge coverage. Additionally, Jiao et al. (2024b) leverage relation graphs derived from web documents to synthesize pretraining data, improving relation-based logical reasoning capabilities.

### 5.1.2 Collecting High-quality Reasoning Trajectories

Beyond constructing high-quality prompts, researchers also refine outputs to collect better trajectories for training. These techniques often sample outputs that follow specific reasoning patterns, such as lengthy reasoning processes with self-reflection, and retain those that meet higher quality standards based on ground-truth labels. Consistent with our architecture definitions in Sec. 2.3, we treat the learned verifier as part of the environment in the agentic system. Consequently, this section focuses exclusively on methods that

utilize existing ground-truth labels—such as answer labels in maths or test cases for code generation—while deferring discussion of methodologies that rely on learned verifiers (reward models or LLM-judges) to Sec. 5.2.

**Rejection sampling**   Rejection sampling (Dong et al., 2023) aims to select higher-quality samples by repeatedly sampling from the policy model (reasoner). Quality is determined through two primary sources: (1) a learned verifier, which we discuss in Section 5.2, and (2) direct comparison with ground-truth labels (when available), where samples inconsistent with the ground-truth labels are discarded. Yuan et al. (2023) apply this idea to mathematical reasoning, introducing edit distance to ensure diversity among trajectories. Zelikman et al. (2022) propose STaR to incorporate the correct answer into the instruction, prompting LLMs to iteratively refine incorrect reasoning traces and generate higher-quality trajectories. Tong et al. (2024) employ an up-sampling strategy to increase the proportion of successful trajectories for more challenging questions. This approach has become a standard technique for iterative model self-improvement, as demonstrated in works such as (Jiao et al., 2025; Guan et al., 2025; Dou et al., 2024b).

**Encourage special reasoning pattern**   Another line of research focuses on leveraging human-like reasoning behaviors—such as self-reflection, deep reasoning, and thinking-before-action—to improve reasoning accuracy and reduce hallucinations. One notable approach is Reasoning-as-Planning (RAP) (Hao et al., 2023), which divides reasoning into three steps: thinking, taking action, and observing (inferring) changes in the environment. When applied to text-based reasoning problems, LLMs simulate environment states after taking actions, leading to more accurate reasoning. Building on this idea, Yuan et al. (2024a) and Chen et al. (2023a) use frontier LLMs like GPT-3.5 and GPT-4 to synthesize trajectories with this pattern for reasoning problems, facilitating imitation learning.

Besides, inspired by the success of long and deep reasoning revealed by OpenAI's o1 model, which incorporate self-reflection and search, some researchers propose imitating this process through rule-based synthesis. For instance, Qin et al. (2024) flatten MCTS trajectories, including failed branches, and ask general models to generate bridge sentences for natural transition from the failed nodes to the ones along the successful paths.

**Reasoning distillation**   Several studies distill reasoning patterns from models capable of producing good reasoning chains (e.g., OpenAI o1) to replicate similar behaviors in smaller models. For example, Huang et al. (2024d), NovaSky Team (2025), Bespoke Labs (2025) and Muennighoff et al. (2025) distill reasoning chains from models like OpenAI-o1, Qwen-QWQ-32B, DeepSeek-R1, and Gemini Thinking Experimental, respectively. Min et al. (2024) diversify this approach by distilling from multiple reasoning models and aggregating outputs into a unified format.

### 5.1.3   Training from Trajectories

Using the collected trajectories, training can be conducted by designing the input and output formats for the algorithms discussed in Section 4.

**Supervised Fine-Tuning (SFT)**   As discussed in Sec. 4.1.1, the most straightforward approach to training reasoning-capable LLMs is to fine-tune a model using SFT on collected trajectories. Methods such as (NovaSky Team, 2025; Bespoke Labs, 2025; Huang et al., 2024d) and (Min et al., 2024) utilize SFT with a modest number of data samples (4K–20K) to replicate the reasoning capabilities of OpenAI's o1 model. Recent SFT approaches have shifted focus to data scaling, with Xu et al. (2025e) exploring the impact of increasing data quantity up to 1 million CoT samples. Their findings demonstrate that performance improves with data scale, albeit with diminishing returns. In contrast, Muennighoff et al. (2025) adopt a sample-efficient approach, curating a high-quality 1K-sample reasoning dataset for fine-tuning. They show that this smaller dataset, combined with strategic inference-time prompting, achieves performance comparable to models trained on larger datasets. Similar strategies have been applied in domain-specific reasoning models, such as earlier math reasoning systems Yu et al. (2023a); Yue et al. (2023).

**Preference learning and reinforcement learning**   While SFT approaches have shown effectiveness, other studies demonstrate that preference learning further enhances performance. Min et al. (2024) study DPO, while Xu et al. (2025e) explore various post-training preference learning methods. Hui et al. (2024),

Min et al. (2024), and Jiao et al. (2024a) all employ DPO with preference pairs derived from code test cases, outcome correctness, and a PRM trained on automatic supervision, respectively. Another line of work focuses on step-level DPO to optimize reasoning action selection. Specifically, Zhang et al. (2024h) use Tree-of-Thought (Yao et al., 2023a) to estimate outcome rewards and backpropagate them to intermediate nodes for quality assessment. Step-level DPO is then applied to pairs sharing the same trajectory prefix but with contrasting next actions. Lai et al. (2024) directly use GPT-4o to identify the earliest incorrect reasoning step and construct contrastive step-level DPO pairs for preference learning. Yuan et al. (2024d) adopt an iterative DPO approach in a self-rewarding setting, where the policy model itself acts as an LLM-as-judge to progressively improve its capabilities.

In addition to preference learning, RL with verifiable answer labels also demonstrate importance in improving reasoning, where rule-based rewards by checking the correctness of sampled solutions are employed rather than reward models.[6] Lambert et al. (2024) use both math reasoning and instruction following data for outcome-based reinforcement learning[7] without reward models. Deepseek-R1 (DeepSeek-AI et al., 2025) further reveal the potential of pure reinforcement learning with verifiable answers. Yu et al. (2025) provide valuable reproduction of Deepseek-R1 on Qwen2.5-32B, including open-sourced data, code, and technical details about loss function design, reward shaping, and dynamic sampling.

**Training with latent reasoning** Typical reasoning models generate long reasoning chains and have demonstrated strong empirical performance. However, this comes at the cost of increased inference time, as they produce lengthy natural language reasoning traces. These traces often contain many tokens that improve the flow and coherence of the output, with only a small fraction directly contributing to the reasoning process. To address this inefficiency, an alternative approach, known as *latent reasoning*, focuses on representing reasoning trajectories implicitly. This is achieved either by omitting intermediate reasoning tokens entirely or by compressing them into specialized reasoning tokens or continuous vector representations.

Earlier work in continuous reasoning focused on compressing natural language reasoning chains into a smaller number of tokens. Deng et al. (2023b) employ knowledge distillation to encode the knowledge from natural language reasoning tokens into intermediate representations of the student model. During inference, the model generates only the final answer without producing additional rationale. This approach is further refined through curriculum learning (Deng et al., 2024b), which gradually removes reasoning tokens during training to reduce distribution mismatch.

However, removing all explicit intermediate reasoning tokens may compromise the model's expressivity (i.e., ability to articulate complex reasoning) (Prystawski et al., 2023). A natural trade-off is to retain a limited number of reasoning tokens, making them implicit to enhance expressiveness while preserving performance. Goyal et al. (2024) introduce learnable `<pause>` tokens during pre-training and fine-tuning within standard CoT trajectories, enabling the model to perform additional computation before generating an output token. Wang et al. (2023d) explore various techniques for compressing reasoning steps from training trajectories into a fixed set of planning tokens. At the start of each reasoning step, the model generates a planning token, whose encoded "knowledge" guides the generation of more coherent outputs. Hao et al. (2024b) propose using the last-layer hidden states before the language modeling head as implicit reasoning token representations, feeding these back into the model to generate the next token auto-regressively. These implicit representations are optimized in a stage-wise manner, akin to the approach of Deng et al. (2024b). Xu et al. (2025f) propose an approach for continuous-space reasoning that does not require modifying the LLM reasoner. Specifically, they employ a lightweight fixed assistant model to generate instance-specific soft thought tokens speculatively as the initial chain of thoughts, which are then mapped into the LLM's representation space via a trainable projection module.

## 5.2   Learning to Reason with Single-agent Systems

As discussed in Section 2.3, agentic systems enhance the reasoning capabilities of standalone LLMs by incorporating agent-environment interactions. These interactions enable the agent to *perceive its environment*

---

[6]We treat the work using reward model/tool-based verifier for RL in the scope of single-agent systems (see Sec. 5.2)

[7]As discussed in Section 4.2, in outcome-based RL, the reward is assigned to the entire trajectory. This contrasts with process-based RL, which assigns a reward at each step.

| Perspective | Method | Characteristic | Representative Work |
|---|---|---|---|
| Action-Environment Interactions | Incorporating Feedback | Use environment feedback to filter trajectories | Ni et al. (2024); Xin et al. (2024b) |
| | Training External Models | Train models (e.g., to critic) from the interaction | Wu et al. (2024c) |
| | Search with Verifiers | Use verifiers to identify better reasoning trajectories | Wan et al. (2024c) |
| | Distillation from Teacher | Distill capability from frontier reasoning model | Gou et al. (2024); Ma et al. (2024a) |
| Training from Trajectories | Supervised Fine-Tuning | Collected offline trajectories + learn via SFT | Dou et al. (2024b); Yin et al. (2024) |
| | Reinforcement Learning | Learning directly on questions and their rewards | Shao et al. (2024) |
| | Learning with Refiner | Train refiner model to iteratively improve the last-round solution. | Xiong et al. (2025) |

Table 7: Summary of learning to reason with single-agent systems.

and accordingly *perform actions*. This section explores how simulation is achieved through the design of such perceptions and agent actions. It then covers training methods—how agents are trained using these trajectories. Additionally, we discuss how predefined patterns are leveraged when collecting trajectories.

### 5.2.1 Trajectory Collection through Agent-Environment Interactions

By interacting with the external world in different ways, agents can effectively construct trajectories that help refine their reasoning process. These interactions to enrich reasoning take the form of (a) incorporating execution feedback, (b) training external models to help reasoning, (c) search with verifiers, and (d) trajectory distillation from stronger teacher agents.

**Incorporating execution feedback**  Through active interaction with the environment, the agent can obtain valuable feedback for trajectory filtering. Building on STaR (Zelikman et al., 2022) (discussed in Sec. 5.1.2), NExT (Ni et al., 2024) leverages unit tests (Ye et al., 2022) to obtain self-generated rationales that lead to correct solutions for training. AlphaProof (AlphaProof & teams, 2024) and DeepSeek-Prover (Xin et al., 2024a) solve formal theorem-proving problems by generating potential solutions and validating them through interaction with the Lean proof assistant (De Moura et al., 2015), either proving or disproving the solutions. Xin et al. (2024b) further improve DeepSeek-Prover by introducing RMaxTS, an exploration strategy driven by intrinsic rewards to generate diverse proof paths. Furthermore, the agent can integrate environmental information directly into the training process to improve its reasoning capabilities. For example, Cummins et al. (2023) train a 7B model from scratch, achieving significantly improved code optimization performance by leveraging optimizing transformations from external LLVM compilers.

**Training external models**  The agent can leverage its interaction with the environment to train external models that can in turn help the agent's reasoning. For example, Wu et al. (2024c) train a critic model to identify relatively easier problems for the policy to explore and guide the policy in searching for deeper proof paths. Re-ReST (Dou et al., 2024b) proposes training a refiner to correct the agent's wrong output based on environmental feedback.

**Reasoning search with verifiers**  Search-based methods address sampling challenges for more difficult problems by leveraging external reward models or generation probabilities to guide decoding. For example, Wan et al. (2024c) develop a Monte Carlo Tree Search (MCTS)-based approach to identify better reasoning trajectories. Each tree node represents either a sentence or token, and a learned LLM-based value function and outcome reward model are used to estimate expected returns during the search process. This method can be applied for both inference-time path selection and training-time imitation learning.

Guan et al. (2025) rely solely on outcome labels to iteratively update the policy model and a process preference model (PPM) through MCTS. The PPM approximates the Q-value of intermediate reasoning steps. Lai et al. (2024) use an LLM-as-judge to identify the first reasoning step in a sampled trajectory that contains an error. The trajectory up to the error is then used to sample new outputs, and DPO preference pairs are formed from correct and incorrect outputs. Zhang et al. (2024h) focus on unsupervised settings where answer labels are unavailable. Discarded steps collected during the search process are treated as negative actions, contrasting with the steps retained in the final path for DPO training. For multi-step reasoning in dynamic environments, such as web navigation, Putta et al. (2024) propose combining guided MCTS with self-critique to facilitate more effective exploration.

**Trajectory distillation from stronger teacher agents**   To tackle challenging mathematical problems, Gou et al. (2024) curate interactive tool-use (e.g., code execution) trajectories using GPT-4, derived from existing mathematical datasets across various domains. Similarly, MuMath-Code (Yin et al., 2024) employs multi-perspective data augmentation to generate diverse math questions and synthesizes code-nested solutions using GPT-4. Beyond mathematics, other domains have also been explored. For instance, Ma et al. (2024a) construct a tool-augmented training set for scientific reasoning by prompting GPT-4. CoGEX (Weir et al., 2024) extends LLMs' program synthesis capabilities to tasks that are not easily expressible as code, such as commonsense reasoning and sarcasm understanding. To collect training trajectories, GPT-4 is used to transform the Alpaca dataset (Taori et al., 2023) into the required format. Ke et al. (2025b) explore collecting trajectories from a more capable generative reward model (GPT-4o) to train a finance-expert model by identifying and correcting the first erroneous step in the reasoning process. Additionally, AgentBank (Song et al., 2024) introduces the largest dataset of agent-environment interaction trajectories, comprising 16 tasks across 5 distinct agent skill dimensions. This dataset is created by annotating actions and their corresponding rationales using LLMs of varying scales, addressing key challenges in trajectory collection, such as scalability.

In addition to leveraging trajectories from GPT-4, Gou et al. (2024) introduce output space shaping by incorporating samples generated by the agent itself. Specifically, they train the agent on both self-sampled correct trajectories and those corrected by a teacher model, promoting diversity in plausible reasoning steps.

### 5.2.2   Agent Training from Trajectories

**Supervised Fine-Tuning (SFT)**   After collecting trajectories, many methods apply supervised fine-tuning (SFT) to train the agent, enabling models with little prior experience in agentic environments to adapt quickly. Dou et al. (2024b) enhances agent reasoning by incorporating refiner-corrected samples into the self-training process. NExT (Ni et al., 2024) uses filtered trajectories to train agents for program repair tasks, while Weir et al. (2024) fine-tune agents on collected trajectories to enable the generation and emulation of pseudo-programs. AlphaProof (AlphaProof & teams, 2024) and DeepSeek-Prover (Xin et al., 2024a) iteratively train and refine the policy model using verified proofs, improving performance in theorem proving tasks. Similarly, Gou et al. (2024), Yin et al. (2024), Ma et al. (2024a), and Song et al. (2024) fine-tune agents on agent-environment interaction trajectories generated by proprietary LLMs, enhancing reasoning capabilities across diverse domains. Notably, MuMath-Code (Yin et al., 2024) adopts a two-stage training strategy, first fine-tuning on pure CoT data and then on code-nested data. Chen et al. (2024e) introduce Agent-FLAN, a fine-tuning method designed to improve LLMs' agent capabilities while addressing challenges such as distribution shifts and hallucinations in training data. By redesigning the training corpus and incorporating negative samples, Agent-FLAN enhances both agent-specific and general capabilities of LLMs.

**Reinforcement Learning (RL)**   Beyond imitation learning through SFT, recent approaches have leveraged reinforcement learning to further enhance reasoning capabilities. Notably, GRPO (Shao et al., 2024; DeepSeek-AI et al., 2025), which employs verifiable outcome rewards during online RL training, has demonstrated strong empirical performance. Havrilla et al. (2024) investigate multiple RL algorithms (e.g., Expert Iteration, PPO) for math reasoning tasks, finding that incorporating outcome reward models has negligible effects on performance for both Expert Iteration and PPO. Similarly, Shao et al. (2024) observe relatively minor performance gains when using PRMs during GRPO training. Yang et al. (2024b) explore using a PRM to "shape" outcome rewards by using a linear combination of outcome and PRM rewards for GRPO training. In contrast, Wang et al. (2024g); Luo et al. (2023a); Jiao et al. (2024a) demonstrate that using a trained PRM during PPO training leads to significant performance improvements. Similar gains are observed in the code generation domain (Dai et al., 2024), where the PRM serves both as a reward signal and as an initial checkpoint for the value function during PPO. Zhang et al. (2024a) iteratively train both a PRM and LLM, while Setlur et al. (2024b) provide a new perspective by comparing Q-value-based PRMs with advantage function-based ones, showing improved learning efficiency and performance in guided reinforcement learning. Concurrently, Gao et al. (2024a) address reward hacking (Casper et al., 2023)—where the policy model generates numerous correct but irrelevant reasoning steps to inflate rewards—by implementing clipping and computing relative, step-adjacent rewards.

| Perspective | Method | Characteristic | Representative Work |
|---|---|---|---|
| Designing Communication | Centralized communication | Use a centralized controller for information aggregation | Canese et al. (2021); Matta et al. (2019) |
| | Conditioned information sharing | Share information based on relevancy and privacy | Hong et al. (2023); Qiu et al. (2024) |
| Coordinating Actions | Leverage knowledge | Utilize expert knowledge as constraints | Lau et al. (2012) |
| | Graph-based methods | Use graphs as structured frameworks | Ruan et al. (2022); Li et al. (2020) |
| | Hiearchical approach | Divide policies to strategy and execution | Xu et al. (2023) |
| Training from Trajectories | Training data from interactions | Obtain high-quality trajectories from interactions | Li et al. (2024c); Estornell et al. (2024) |
| | Gradient modification | Modify gradients towards optimal points | Li et al. (2024f) |

Table 8: Summary of learning to reason for multi-agent systems.

Qiao et al. (2023a) introduce TRICE, a two-stage framework that enables agents to determine when and how to use tools through Reinforcement Learning with Execution Feedback (RLEF) from external tools. Similarly, Xin et al. (2024b) enhance DeepSeek-Prover by incorporating reinforcement learning from proof assistant feedback (RLPAF). To effectively learn from both successful and unsuccessful agent-environment interactions, Putta et al. (2024) develop an off-policy variant of DPO for iterative training.

**Learning with refiner** For more challenging questions, models may fail to generate enough successful trajectories to serve as a reliable positive training signal. However, even trajectories with incorrect outcomes can still be leveraged effectively. For example, Qu et al. (2024a) train a correction model using RL to iteratively refine generated model responses. Similarly, Tang et al. (2025) propose a self-evolving framework to train a critique model, which enhances the quality of outputs through continuous feedback.

Refiner models can also be integrated into the search process to iteratively improve generation quality. For instance, Snell et al. (2024) train a refiner model via RL (Qu et al., 2024b) to refine outputs sequentially. The final prediction is obtained through majority voting over all predictions generated during this iterative refinement process, effectively scaling test-time computation. Xi et al. (2024) develop a step-level critique model that provides feedback for each reasoning step, using training instances collected from GPT-4o. This feedback serves two purposes: (1) expanding training data to improve the actor model, and (2) scaling test-time computation through iterative self-refinement in a multi-agent setup. Zhang et al. (2024b) combine reasoning and self-refinement into a single MCTS framework, where each node is either a reasoning node (generating complete reasoning trajectories) or a refining node (identifying and correcting reasoning flaws). A learned pairwise reward model compares the quality of refined and original outputs, estimating the expected returns of each node. However, this work does not explicitly account for the inference setting, where neither the reasoner nor the refiner has access to the correctness of the sampled response. This can lead to refiners inadvertently degrading originally correct solutions. To address this issue, Xiong et al. (2025) introduce a learnable self-rewarding mechanism. This approach mitigates the risk of worsening correct solutions and alleviates the distribution-shifting problem in self-correction (Kumar et al., 2024).

## 5.3 Learning to Reason with Multi-agent System

In Section 2.3, we discussed how multi-agent systems extend single-agent systems through agent-agent communication. This enables agents to assume distinct roles, exchange messages, and coordinate their actions before interacting with the environment. In this section, we explore how trajectory collection can be achieved through the careful design of agent-agent *communication* and the coordination of *actions* across different agents. As a system level, communication serves as the input or perception mechanism for participating agents, focusing on the protocols governing message exchange. Meanwhile, actions represent the output of the system, addressing how consensus is reached given the diverse actions proposed by individual agents.

### 5.3.1 Designing Agent-Agent Communication

In a multi-agent framework, ensuring that each agent is aware of the actions of others is critical, as a well-designed communication system can significantly enhance collective intelligence (Guo et al., 2024b). One effective solution is the use of a centralized controller (Canese et al., 2021). For example, Matta et al. (2019) propose a centralized aggregation center that constructs a global swarm matrix by aggregating the Q-value tables of all agents. Similarly, the MARCO framework (Zhang et al., 2021) employs centralized training with

decentralized execution to improve sample efficiency in partially observable multi-agent environments. By learning a shared model that generalizes across agents' policies and directing exploration toward uncertain areas, MARCO optimizes reasoning and resource utilization in cooperative tasks.

To enable effective communication among agents, Sukhbaatar et al. (2016) introduce a neural communication model with a learned protocol tailored to the task. Additionally, a shared message pool (Hong et al., 2023) can be implemented, where agents send messages and subscribe to relevant ones based on their individual profiles. In recent work by Qiu et al. (2024), each agent maintains a private intention, which includes its current goal and associated sub-tasks. These intentions are broadcast periodically, and a propagation network converts them into teammate-specific communication messages, ensuring that relevant goals are shared with the appropriate teammates.

### 5.3.2 Coordinating Actions among Multiple Agents

To enhance coordination among multiple agents, various approaches have been proposed, including leveraging expert knowledge, graph-based frameworks, and hierarchical structures to improve efficiency and effectiveness. For better coordination of actions across agents, Lau et al. (2012) utilize expert coordination knowledge as constraints to refine the exploration and learning process. By reducing the action space and focusing on promising states, this approach enhances decision-making. Additionally, graph-based methods have been explored to improve coordination. For instance, the Graph-based Coordination Strategy (GCS) (Ruan et al., 2022) introduces a framework that employs a directed acyclic graph to coordinate agent policies. This enables agents to synchronize their actions through predefined temporal sequences. Similarly, Deep Implicit Coordination Graphs (DICG) (Li et al., 2020) propose a graph neural network-based module to dynamically infer coordination structures for multi-agent reinforcement learning (MARL).

Furthermore, hierarchical approaches have been developed to enhance synchronization. The Hierarchical Cooperative Multi-Agent Learning (HAVEN) framework (Xu et al., 2023) divides policies into two levels—strategy and execution—improving both inter-agent and inter-level coordination.

### 5.3.3 Multi-Agent Training from Trajectories

Compared to single-agent scenarios, multi-agent training introduces additional challenges in higher coordination and communication complexity and recent approaches have leveraged different ways to address the challenge. DEBATUNE (Li et al., 2024c) employs a multi-round debate mechanism between two agents with opposing stances to generate training data. Through iterative debate, arguments are refined, resulting in high-quality and diverse outputs. During the training phase, models are fine-tuned using these debate-generated trajectories, enabling controllability and alignment with user-defined stances. Similarly, Subramaniam et al. (2025) fine-tune a society of agents, starting from the same base model, on independent data generated through multi-agent interactions. These agents specialize in distinct roles, such as "generation" and "critic" producing diverse reasoning trajectories. Training on such varied trajectories fosters specialization and mitigates performance plateaus. Acc-Debate (Estornell et al., 2024) utilizes an Actor-Critic framework to train a team of two agents collaboratively. One agent serves as the "Actor" generating responses, while the other acts as the "Critic" refining those responses. Training alternates between optimizing the Actor and Critic models, leveraging partial trajectory rewards which captures the expectation of reaching the correct answer at intermediate time stepsto address temporal dependencies in the debate process. This approach enhances collaboration and improves final performance.

Furthermore, Li et al. (2024f) address the challenge of mixed-motive cooperation in multi-agent systems by modifying gradients to guide agents toward stable fixed points that balance individual and collective interests. This method enhances the ability to optimize trajectories for effective collaboration.

## 5.4 Toward Cost-aware and Inference-aware Training

As reasoning models grow increasingly complex, ensuring both efficiency and effectiveness becomes crucial. Inference-time scaling and learning-to-reason approaches play complementary roles, as most inference-time scaling methods can be applied to models specifically trained for reasoning. However, both approaches come

with associated costs, whether it involves generating thousands of additional tokens compared to greedy decoding during inference or training models on large-scale trajectory datasets. Consequently, *cost-aware* methodologies, which factor in computational costs when deciding how to allocate resources during both training and inference, or those that address sample inefficiency, have gained recent attention. Similarly, *inference-aware* methodologies aim to enhance the time and cost efficiency of inference scaling by explicitly incorporating inference-time scaling strategies during training. In this section, we explore emerging cost-aware and inference-aware approaches.

### 5.4.1 Cost-aware Training

**Learning to reduce inference cost** This line of research explores strategies to optimize the trade-off between computational cost and reasoning performance by dynamically allocating resources based on input (prompt) complexity and desired output quality. For prompt analysis, Damani et al. (2025) use a learnable model to predict the difficulty of batched queries and dynamically allocate inference budgets accordingly. Building on this, Zhang et al. (2024d) train a model to predict the most efficient combination of inference strategies, directly optimizing for pass rates. Yue et al. (2025) decompose reasoning trajectories into specific behaviors and employ a trainable planner to derive question-specific compositions, identifying the optimal reasoning strategy—such as whether question decomposition or rewriting is necessary, whether Python programs are required, or if answer verification is needed. On the output side, Snell et al. (2025) propose a look-ahead search method, similar to step-level beam search, which switches between branches based on estimated returns to minimize search costs.

**Data-efficient training** Another research direction focuses on reducing training costs by using a small set of high-quality samples (questions paired with trajectories or labels). Muennighoff et al. (2025) curate a dataset of 1,000 samples, emphasizing difficulty, diversity, and quality. Their work demonstrates that fine-tuning Qwen2.5-32B-Instruct on this dataset achieves performance surpassing o1-preview on competition math benchmarks. Ye et al. (2025) fine-tune Qwen2.5-32B-Instruct on 817 carefully curated training samples, achieving superior performance across a broader set of math reasoning benchmarks. Notably, Ye et al. (2025) highlight that these performance gains depend on using strong pre-trained models like Qwen2.5-32B-Instruct and do not occur with weaker models (*e.g.*, Qwen1.5-32B-Instruct).

### 5.4.2 Inference-aware Training

Existing work on inference scaling typically treats inference-time computation as a post-hoc design choice after conventional training. Inference-aware training approach challenges the assumption that decoupling training and inference-time computation is optimal. For instance, if an LLM is allowed multiple attempts to solve a math problem, fine-tuning it to explore diverse problem-solving strategies might yield better results than simply generating candidates representing its best single attempt.

The core idea is that explicitly considering the inference procedure during training can significantly enhance the effectiveness of inference-time computation. For example, Best-of-N (BoN) is a basic inference-time strategy that selects the highest-reward response from $N$ candidates. However, this approach is misaligned with fine-tuning objectives. To address this, Sessa et al. (2024) propose an RL objective that distills the Best-of-N distribution into the policy model using Jeffreys divergence (Jeffreys, 1946). Similarly, Balashankar et al. (2024) develop a calibrated reward that incorporates the inference procedure (Best-of-N) during alignment. In a related effort, Chow et al. (2024) aim to optimize BoN directly, overcoming the non-differentiable argmax operator by employing a reinforcement learning framework.

## 6 Discussion: Trends and Open Challenges

The field of reasoning LLMs has seen rapid advancements, with notable trends emerging in training-vs-inference regimes and architectural dimensions as we discuss in Section 6.1. Despite this progress, several challenges remain, hindering their generalizability and practical applicability. This section outlines these observed trends and highlights open challenges, along with potential directions to address them (Section 6.2).

## 6.1 Observed Trends

Following the two dimensions outlined in Figure 2, we identify two key trends in LLM reasoning: one progresses from inference scaling to learning to reason (Section 6.1.1), while the other shifts from standalone LLMs to agentic systems (Section 6.1.2). Additionally, reasoning is ubiquitous yet challenging when developing a general-purpose reasoner. Notably, many state-of-the-art reasoning language models are predominantly focused on a few domains, particularly mathematics and coding (OpenAI et al., 2024; DeepSeek-AI et al., 2025). Whether it is possible to build a truly generalizable reasoning system remains an open question (Kang et al., 2024; Qi et al., 2024; Huang et al., 2024c; Sun et al., 2024c). However, we observe a growing trend toward developing domain-specific reasoning models (Section 6.1.3).

### 6.1.1 From Inference Scaling to Learning to Reason

Since the introduction of CoT and self-consistency (Wang et al., 2023f), inference scaling techniques have emerged as a key paradigm for enhancing reasoning performance without incurring the costs associated with reasoning-specific training. Inference scaling complements learning-to-reason approaches, with recent studies demonstrating that combining self-consistency with reasoning-specific training yields further improvements (DeepSeek-AI et al., 2025; Muennighoff et al., 2025). Additionally, since the release of OpenAI's o1 (Huang et al., 2024d), some methods have sought to activate human-like reasoning patterns by introducing self-correction (Kumar et al., 2024), self-critique (Xi et al., 2024), or even MCTS Qin et al. (2024).

Researchers initially found that data-driven approaches, such as supervised fine-tuning (SFT) and knowledge distillation, were highly effective in enhancing LLMs' reasoning capabilities. However, these methods rely on the availability of a strong teacher model for distillation. An alternative approach uses outcome labels for iterative rejection sampling (Yuan et al., 2023), which converges quickly after a few iterations (Dong et al., 2023). These limitations have spurred the development of more data-efficient methods, such as automatic process supervision (Jiao et al., 2024a; Wang et al., 2024g;k; Luo et al., 2024b) and iterative refinement (Guan et al., 2025), which optimize training trajectories using fixed outcome labels. The release of Deepseek-R1 (DeepSeek-AI et al., 2025) further advanced the field, demonstrating the ability to generate human-like, long reasoning chains through pure reinforcement learning under outcome supervision alone.

### 6.1.2 From Standalone LLMs to Agentic Systems

In Sections 2.3 and 5, we discussed how the rise of agentic systems has significantly influenced reasoning research. A clear trend has emerged, shifting from standalone LLM reasoning to agentic reasoning. This shift aligns with our expectations: reasoning is no longer confined to a single LLM but is expected to interact with the external world and other agents, as well as exhibit autonomy, such as planning capabilities.

On one hand, there is ongoing debate about whether agentic reasoning is *always* beneficial, especially for straightforward and simple tasks (Sprague et al., 2024b; Liu et al., 2024c). On the other hand, current systems' *autonomy* is largely limited to *planning*, whereas it could encompass much more. For instance, *system-level or meta-level planning* is essential in agentic systems, requiring the design of effective ways to connect different agents (Zhou et al., 2025a; Zhuge et al., 2024; Zhang et al., 2024c; Hu et al., 2025). A notable recent study (Ke et al., 2025c) demonstrates that such design can be with *zero supervision* and through self-improvement alone. Another critical aspect of autonomous agents is *proactiveness*, yet current reasoning agents still lack the ability to proactively seek clarification or request additional information from users or the environment.

### 6.1.3 Domain-Specific Reasoners

**Mathematical reasoning**   Mathematics serves as an ideal testbed for studying LLM reasoning capabilities due to its structured nature and clear evaluation criteria. Mathematical reasoning has evolved along two complementary paths. The first, often referred to as the "informal approach" (Yang et al., 2024d), treats mathematical problems as natural language tasks and fine-tunes LLMs on carefully curated or filtered problem-solving datasets. Systems like NuminaMath (Fleureau et al., 2024), DeepSeekMath (Shao et al., 2024), Llemma (Azerbayev et al., 2024), and MetaMath (Yu et al., 2024b) have demonstrated remarkable

capabilities by combining mathematical text training (pre-training, supervised fine-tuning, and reinforcement learning), tree-based search, tool-integrated reasoning, and various inference scaling techniques discussed in earlier sections. This approach has achieved significant success across benchmarks ranging from GSM8K (Cobbe et al., 2021) and MATH (Hendrycks et al., 2021b) to competition-level problems such as AIMO (Markets, 2024) and AIME-level problems (aim, 2025). However, challenges persist in tackling college-level and advanced mathematics, where high-quality training data is scarce, and verifying complex multi-step reasoning becomes increasingly difficult. Spatial reasoning (*e.g.,* counting, navigation, and inferring spatial relationships) presents another challenge for LLMs and multi-modal LLMs (Wang et al., 2024b).

Complementing the informal approach, formal mathematical reasoning grounds systems in precise symbolic frameworks, such as proof assistants like Isabelle (Nipkow et al., 2002), Lean (De Moura et al., 2015), and Coq (Barras et al., 1997; The Coq Development Team, 2024). Recent advances in this direction include neural theorem-proving systems that combine tactic generation with proof search (Yang et al., 2023b; Thakur et al., 2024), as well as autoformalization techniques that translate between natural and formal mathematics (Wu et al., 2022; Jiang et al., 2024a). The formal approach offers several advantages: automatic verification of reasoning steps, generation of training signals from the verification environment, and the potential to bootstrap capabilities through learned abstractions. For example, AlphaProof (AlphaProof & teams, 2024) and AlphaGeometry (Trinh et al., 2024) demonstrate the power of integrating neural networks with symbolic verification, achieving groundbreaking performance on Olympic-level mathematics problems. A recent position paper by Yang et al. (2024d) argues that formal mathematical reasoning represents a critical frontier for advancing AI's ability to tackle increasingly abstract and complex mathematical problems.

**Code generation**   Code serves as a more formal language for reasoning. Given the complexity of generating entire programs, earlier studies primarily focused on function-level code completion, as demonstrated by benchmarks such as HumanEval (Chen et al., 2021) and MBPP (Austin et al., 2021). With stronger foundation models trained on extensive code corpora (Zhu et al., 2024a; Hui et al., 2024), the focus of evaluation has shifted toward general competition programming (Hendrycks et al., 2021a; Jain et al., 2024). The earliest significant attempt to solve competition-level coding problems through large-scale training was AlphaCode (Li et al., 2022). Similar to the general domain, the training paradigm has evolved from instruction tuning (Wei et al., 2024) to RL and preference learning based on test cases and compiler feedback (Dou et al., 2024a; Weyssow et al., 2024; Jiao et al., 2025; Huang et al., 2024b). The recent releases of DeepSeek-R1 (DeepSeek-AI et al., 2025) and OpenAI's o3 (OpenAI et al., 2025) have further advanced the field by enabling end-to-end RL through outcome supervision. OpenAI et al. (2025) also highlight that purely data-driven approaches can outperform models incorporating human-experience-based competition strategies.

Another important application of code generation is in software engineering, where advancements in LLMs are making fully automated pipelines increasingly feasible. SWE-Bench (Jimenez et al., 2024), a benchmark based on GitHub issues, challenges LLMs with real-world software engineering problems. These tasks require coupled abilities, such as long-context modeling to process repository-level inputs, logical reasoning to locate bugs and design unit tests, and programming to implement solutions. Wei et al. (2025) pioneer the use of end-to-end RL for optimizing automatic debugging. Specifically, they select pull requests (PRs) from GitHub linked to issues and use the consistency between the predicted code snippet and the repository's code after the PR is merged as the reward signal.

**Tabular reasoning**   Reasoning over tabular (or structured) data, which involves generating responses based on user queries and provided tables, plays a vital role in improving data analysis efficiency (Lu et al., 2025). A critical aspect of tabular reasoning with LLMs involves transforming structured data into a format that these models can process effectively. Techniques such as serialization (Chen, 2023; Cheng et al., 2023; Chen et al., 2023e), prompt engineering (Ye et al., 2023b; Lin et al., 2023b; Wang et al., 2024n; Zhang et al., 2024j), and embedding methods (Herzig et al., 2020) have been widely studied to facilitate this adaptation, converting tabular data into human-readable text or leveraging specialized table representations. Additionally, specialized prompting of LLMs with transformed tabular data is crucial. For instance, Pourreza & Rafiei (2023); Ye et al. (2023c) find that LLMs perform better on decomposed sub-tasks than on the entire table reasoning task. However, LLMs may still struggle with certain sub-tasks. To address this, (Cao et al., 2023) employ diverse tools for specific sub-tasks, while (Lin et al., 2023b;a) focus on retrieving relevant

tables. Notably, (Jiang et al., 2023) propose a unified approach to enhance LLM reasoning over structured data by designing specialized interfaces. These interfaces extract relevant evidence from structured data, enabling LLMs to focus on reasoning based on the gathered information.

Despite the promising results of various adaptation methods, significant challenges remain. First, tabular data often comprises diverse feature types—categorical, numerical, and textual—adding complexity to modeling (Borisov et al., 2023; Gruver et al., 2023). Second, the effectiveness (Sui et al., 2024) and robustness (Liu et al., 2024d) of LLMs in tabular tasks heavily depend on proper prompt design and data preprocessing. Poor or out-of-distribution preprocessing can lead to information loss, misinterpretation, multicollinearity, and interpretability issues, significantly degrading performance (Sui et al., 2024). Finally, LLMs are prone to hallucinations (Ye et al., 2023d) and fairness concerns (Liu et al., 2023), limiting their reliability. For a comprehensive overview, see recent surveys on LLMs for table reasoning (Fang et al., 2024b; Dong & Wang, 2024; Zhang et al., 2025a; Lu et al., 2025).

**Reasoning in multi-agent games**  In game-theoretic scenarios involving both collaboration and competition, strategic social reasoning skills are essential (Lee et al., 2024). Strategic reasoning refers to the cognitive process of making decisions in complex social situations. As highlighted by Feng et al. (2024b), the complexity and challenges of this reasoning stem from the involvement of multiple parties and the dynamic nature of the environment.

To capture the cognitive states of multiple parties, the concept of Theory-of-Mind (ToM) (Zhang et al., 2012) has been integrated into modeling processes. ToM attributes mental states—such as beliefs, intentions, desires, emotions, and knowledge—to oneself and others. Recent studies (Kosinski, 2024) have shown that LLMs exhibit ToM capabilities, and researchers have leveraged these capabilities to enhance strategic reasoning in social scenarios. For instance, Guo et al. (2023) computationally model the beliefs, intents, and potential behaviors of teammates and opponents to improve understanding and reasoning in games. Similarly, TOMABD (Montes et al., 2023) incorporates ToM into agents to enhance their reasoning and decision-making abilities. To address the complexity of dynamic social interactions (Li et al., 2024d), prior research employs RL methods to explore potential behaviors and evaluate different states (Seo & Lee, 2017; Wen et al., 2019). Additionally, some studies introduce modular frameworks to improve strategic reasoning in complex scenarios. For example, ReTA (Duan et al., 2024) uses LLM-based modules as the main actor, reward actor, and anticipation actor, inspired by minimax game theory. Recent work (Trencsenyi et al., 2025) has also begun exploring role-based multi-agent interactions to enable more sophisticated strategic reasoning. These approaches collectively enhance LLMs' strategic reasoning capabilities in dynamic environments.

**Reward modeling and evaluation as a reasoning task**  Evaluation, whether as an end goal or a component of a larger reasoning system, remains a significant challenge. While using PRMs to enhance reasoning abilities is popular during both inference and training, training these models requires extensive step-by-step annotations (Lightman et al., 2024). To address this, recent approaches have introduced automated feedback mechanisms, such as tree search (Wang et al., 2024g; Chen et al., 2024a; Setlur et al., 2024a; Luo et al., 2024c; Wang et al., 2024l) or, less frequently, LLM-as-judge (Zhang et al., 2025b). Although these methods avoid human preference annotations, they often rely on trajectories sampled from a fixed policy model, which may not align well with the problem distribution. This misalignment leads to poor generalization, as highlighted by Zheng et al. (2024). Consequently, the next frontier in reward modeling will need to combine automated data collection with diverse data sources to achieve annotation-efficient generalization.

While reasoning in LLM-as-judges is not explicitly addressed, recent training and inference techniques have drawn from established methods for improving reasoning. Judge-based assessment inherently involves a finite set of outcomes (e.g., A or B for pairwise judgments or 1-5 for single ratings), making it suitable for self-consistency decoding (Kim et al., 2024b). More advanced inference-time approaches, such as multi-judge or multi-round discussions (Li et al., 2023d; Chan et al., 2023; Verga et al., 2024; Yu et al., 2024d), self-rationalization (Trivedi et al., 2024), or sequential escalation (Jung et al., 2024), have been proposed. Concurrently, training-time solutions for LLM-as-judges focus on distilling chain-of-thought judgments from larger teacher models and fine-tuning smaller judges via supervised fine-tuning (Wang et al., 2023g; Li et al., 2023b; Kim et al., 2023; 2024c; Vu et al., 2024) or preference optimization (Hu et al., 2024; Wang et al.,

2024f; Ye et al., 2024; Saad-Falcon et al., 2024; Deshpande et al., 2024; Wang et al., 2024j). Despite these advancements, such models still struggle in reasoning-intensive domains (Tan et al., 2024; Zhou et al., 2025b; Xu et al., 2025b), whereas stronger reasoning models have outperformed specialized judge models in more difficult evaluation settings (Xu et al., 2025a). In all, recent benchmarking results highlight that developing reasoning-specific judges remains an open and challenging research area.

## 6.2 Open Challenges

Despite the trends observed in Section 6.1, several challenges remain. First, how can we effectively evaluate both the reasoning outcome and the reasoning chain? (Section 6.2.1). Second, do we truly understand reasoning? Does the reasoning chain generated by next-token sampling faithfully reflect the internal reasoning process of an LLM, or is it merely imitating its training data? (Section 6.2.2). Third, training of LLM reasoning system is still largely hindered by substantial data requirements, which include both more challenging questions and the corresponding outcome labels. This not only affects the end-to-end reasoner training, but also limits our exploration in building stronger reward models to facilitate inference time scaling (Section 6.2.3).

### 6.2.1 Evaluating Reasoning

As language models and agentic systems tackle increasingly complex tasks, evaluating their performance becomes equally challenging. Currently, progress in LLM reasoning is measured by *outcome* performance on fixed benchmarks (e.g., MATH (Hendrycks et al., 2021b)). However, relying solely on outcomes to verify reasoning correctness may be insufficient, as a correct final answer does not guarantee a logically sound reasoning chain (Hao et al., 2024a). Prior work has shown that LLMs often produce unfaithful reasoning chains, even when the final answers are correct (Wiegreffe et al., 2022; Lyu et al., 2023; Wang et al., 2023b).

Evaluating reasoning beyond outcomes remains an open and challenging problem. Early approaches relied on human annotators to assess the quality of generated explanations (Camburu et al., 2018; Rajani et al., 2019), focusing on whether the reasoning could lead to the same predictions. To scale this idea, follow-up works (Wiegreffe et al., 2020; Hase et al., 2020) used trained models as simulators to evaluate the alignment between generated reasoning and final predictions. When human-annotated reasoning chains are available, some studies leverage traditional NLG metrics to measure overlap between human- and model-generated explanations (Clinciu et al., 2021). Others propose reasoning-specific metrics to assess aspects like coherency, redundancy, factuality (Golovneva et al., 2022), informativeness (Chen et al., 2022), robustness (Wang & Zhao, 2024), and contextual faithfulness (Ming et al., 2025). Under the LLM-as-Judge paradigm, recent works prompt powerful LLMs like GPT-4 to directly evaluate reasoning chains generated by other models (Hao et al., 2024a; Sun et al., 2024b). However, as reasoning tasks grow in complexity, evaluation becomes increasingly difficult, even for frontier models—if a model cannot perform a task, how can it judge if the task is done correctly? Thus, developing robust and accurate methods to evaluate reasoning beyond outcomes remains a significant and unresolved challenge.

### 6.2.2 Understanding Reasoning

Recent research on understanding LLM reasoning has advanced along two complementary paths: empirical studies that evaluate and analyze performance through carefully designed and controlled experiments, and formal analyses that introduce new frameworks to systematically explore the underlying mechanisms of how LLMs reason.

**Empirical analysis of reasoning** Recent LLMs exhibit strong performance across diverse tasks, suggesting some level of reasoning capability. However, whether these skills are general and transferable or merely specialized for tasks encountered during pretraining remains an open and debated question. To address this, several empirical studies have sought to understand and enhance LLM capabilities across various reasoning forms: abstractive reasoning (Wu et al., 2024a; He & Lu, 2024), compositional reasoning (Bhargava & Ng, 2022; Li et al., 2024g), inductive reasoning (Yang et al., 2024f; Han et al., 2024b), abductive reasoning (Jung et al., 2022; Pareschi, 2023), deductive reasoning (Poesia et al., 2024; Seals & Shalin, 2024; Feng et al.,

2024a), logical reasoning (Wan et al., 2024b; Han et al., 2024a; Xu et al., 2025c), commonsense reasoning (Lin et al., 2021; Liang et al., 2023a; Sun et al., 2024a), math reasoning (Ahn et al., 2024; Mirzadeh et al., 2025), and social reasoning (Gandhi et al., 2023). Notably, Arkoudas (2023) qualitatively evaluate GPT-4 on 21 diverse reasoning problems, concluding that despite occasional analytical success, GPT-4 remains incapable of true reasoning. Similarly, Wu et al. (2024a) empirically investigate abstractive reasoning and find that while LLMs achieve nontrivial performance on counterfactual tasks, their performance consistently degrades compared to default conditions, indicating reliance on narrow, non-transferable procedures. Mondorf & Plank (2024) provide a comprehensive survey on recent evaluations of LLM reasoning abilities.

Beyond assessing LLM reasoning capabilities, there is growing interest in evaluating how test-time scaling methods enhance reasoning. The empirical success of CoT prompting has spurred extensive research into its mechanisms. Wang et al. (2023a) and Madaan et al. (2023a) investigate the role of demonstrations, finding that LLMs prioritize pattern consistency over accuracy and exhibit robustness to invalid demonstrations—particularly in mathematical reasoning, where incorrect equations often do not hinder performance. They also emphasize the importance of relevant rationales and logical progression in CoT prompts. Additionally, Madaan et al. (2023a) conclude that CoT aids models by supplementing missing information, such as commonsense knowledge, and reinforcing task understanding. From a modeling perspective, Dutta et al. (2024) analyze CoT through neural mechanisms, revealing that LLMs process input context and generated CoT via parallel pathways. They find that early layers (e.g., layers 1-16 in Llama-2 7B (Touvron et al., 2023)) rely on pretraining knowledge, while later layers specialize in in-context learning, with answer-writing heads emerging in the final layers. From a task perspective, Sprague et al. (2024a) conduct a meta-analysis of 100 CoT papers, showing that CoT significantly improves performance on mathematical, logical, and algorithmic reasoning tasks but offers minimal gains for non-symbolic tasks. Their analysis suggests that CoT excels in computational steps but struggles with tool-augmented reasoning. On the training front, Gao et al. (2024a); Zhang et al. (2025b); Yeo et al. (2025) explore key supervised fine-tuning (SFT) and reinforcement learning (RL) factors that optimize LLM training strategies for enhancing CoT reasoning.

**Formal analysis of reasoning** There is increasing interest in formal analyses, which use structured and logical proofs to systematically evaluate and improve the reasoning capabilities of LLMs. Han et al. (2022) introduce FOLIO, a dataset designed to assess models' ability to derive correct conclusions from premises using first-order logic reasoning. Similarly, Saparov & He (2023) develop a benchmark evaluating LLMs on symbolic ontologies, revealing that models often struggle with proof planning and rely on knowledge retrieval rather than genuine reasoning. These findings highlight the potential of neurosymbolic methods to better understand LLM reasoning. Recent work also explores formal analysis techniques to enhance LLM reasoning. For instance, Pan et al. (2023) use LLMs to translate natural language problems into symbolic formulations, which are then processed by deterministic symbolic solvers for inference. (Li et al., 2025b) demonstrate the promise of leveraging LLMs' symbolic reasoning for mathematical problem-solving. Other studies focus on domain-specific reasoning: Fang et al. (2024a) propose an LLM-based agent for text-based games, designed to tackle symbolic challenges and achieve in-game objectives, while Nahid & Rafiei (2024) introduce a framework to enhance LLMs' symbolic reasoning by normalizing web tables. These studies reveal LLMs' limitations in structured reasoning while emphasizing the value of integrating formal analysis to strengthen their capabilities.

**Theoretical analysis of ICL and CoT reasoning** The success of in-context learning (ICL) and CoT prompting in enhancing LLM reasoning has sparked significant interest in understanding their underlying mechanisms from theoretical perspectives. Extensive prior studies on ICL suggest that transformer-based in-context learners effectively implement various learning algorithms, encoding implicit, context-dependent models for generation within their hidden activations—models that can be trained through demonstrations as these activations are computed. For instance, Akyürek et al. (2022) investigate this hypothesis in the context of linear regression models, while Von Oswald et al. (2023) and Dai et al. (2023) explore how transformer-based in-context learners function as meta-optimizers, effectively learning models via gradient descent during their forward pass. From a Bayesian inference perspective, Xie et al. (2022); Zhang et al. (2023) and Wang et al. (2023e) demonstrate that transformer-based in-context learners can achieve the Bayes-optimal predictor when demonstrations are selected based on a shared latent concept variable, such

as format or task information, even in the presence of distribution mismatches between demonstrations and training data. Additionally, Elhage et al. (2021); Olsson et al. (2022) examine ICL through the concept of "induction heads" – attention heads that implement a simple algorithm to complete tasks, providing evidence that induction heads may underlie much of the in-context learning observed in transformer-based models.

The body of work exploring the theoretical insights into CoT mechanisms remains relatively limited, with most studies focusing on the expressiveness of LLMs when using CoT. A pioneering study by Feng et al. (2023a) investigates LLMs with CoT for solving mathematical and decision-making problems. Using circuit complexity theory (Arora & Barak, 2009), they demonstrate that bounded-depth transformers cannot solve basic arithmetic or equation tasks unless the model size grows super-polynomially. In contrast, they prove that constant-size models can solve these tasks, along with a wide range of decision-making problems such as Dynamic Programming, by generating CoT derivations in a common mathematical language. Li et al. (2024h) extend these findings, providing a tighter upper bound on the expressiveness of constant-depth transformers with CoT. However, these studies do not explore how the length of a CoT affects model reasoning power. To address this gap, Merrill & Sabharwal (2024) find that a logarithmic number of intermediate steps (relative to input length) offers only marginal gains over standard transformers, while a linear number of steps under the assumption of projected pre-norm (a slight generalization of standard pre-norm) enables the recognition of all regular languages. Furthermore, polynomially many steps, combined with generalized pre-norm, allow transformers to recognize exactly the class of polynomial-time solvable problems.

### 6.2.3 Data Challenges in Advancing Reasoning Capabilities

**Challenges in scaling question and outcome supervision for RL**   As discussed earlier, development trends in both general and task-specific domains are converging, with a focus on employing end-to-end RL to minimize inductive bias and push the boundaries of intelligence. Frontier models now incorporate competition-level problems annually for training, as these represent the most challenging tasks and are annotated with high-quality answers by human experts. However, we are nearing the limits of available human-annotated data, raising the question of whether methods beyond human labeling can enable the continuous scaling of RL. This challenge is particularly relevant in domains where prompts are not easily verifiable, such as open-ended generation, software engineering, and most agentic tasks.

**Challenges in reward modeling**   Early studies have investigated the feasibility of process supervision (Lightman et al., 2024) and its effectiveness in inference-time scaling (Snell et al., 2025). However, its high annotation costs and ambiguous definition—particularly in long CoT scenarios where self-reflection is encouraged—have limited its adoption in large-scale reinforcement learning. Despite these challenges, the key advantage of accurate process supervision is its ability to reduce hallucinations, making it essential for automated reasoning and knowledge discovery. Additionally, as discussed in Section 4.2, the training paradigm for reward models is closely tied to that of reasoning models. This raises concerns about whether allocating the same annotation budget directly to reasoning models could lead to more stable and general improvements, potentially limiting the gains achievable through inference-time scaling.

## 7   Conclusion

In this work, we provide a timely and comprehensive survey on LLM reasoning. We first formalize the goal of LLM reasoning and consolidate past research by categorizing reasoning techniques along two dimensions: regimes and architectures. Within each of these dimensions, we review both input and output perspectives in detail. Our review highlights emerging trends, including the shift from inference-time scaling to learning-to-reason regimes, and the transition from standalone models to agentic systems. We also review and compare a wide range of learning algorithms, including supervised fine-tuning and reinforcement learning, as well as the training of reasoners and training of verifiers. Despite these advancements, challenges remain in evaluating reasoning and understanding real reasoning mechanisms as well as addressing data challenges in advancing reasoning capabilities. We encourage future research to further explore these trends, such as inference-aware learning-to-reason and automated multi-agent design, to enhance LLM reasoning.

**Acknowledgments**

We thank M Saiful Bari, Semih Yavuz and Yingbo Zhou for helpful discussions.

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
