# OpenReview forum: "A Survey of Frontiers in LLM Reasoning: Inference Scaling, Learning to Reason, and Agentic Systems"
_TMLR — Accepted by TMLR_

### Review · Reviewer_fmKu · 2025-04-09

**Summary Of Contributions:**

This paper presents a comprehensive survey on the current frontier topics of LLM reasoning, including test-time scaling, (long) reasoning, and agents. The existing methods are evaluated in several dimensions of taxonomy, including training stage, the complexity of LLM system (standalone / agent / multi-agent), and input / output optimization. Under such taxonomy, the paper introduces inference scaling techniques and learning-based works for LLM reasoning. The paper also discusses the trends and open challenges of LLM reasoning, pointing out that LLMs starts to shift from inference scaling to learning to reason, from standalone LLMs to agentic systems, and domain-specific models, and faces the challenge of evaluation, understanding reasoning and limited data.

**Audience:**

Yes

**Broader Impact Concerns:**

This paper does not contain a broader impact statement section. That being said, I think this paper does not have significant concerns on the ethical implications of the work as it is a survey without any experiment or proposed methodology.

**Claims And Evidence:**

Yes

**Requested Changes:**

**Major Changes**

1. In the caption of Fig. 1, the authors mention that "research on regimes and architectures has accelerated notably since the introduction of Chain-of-Thought (CoT) in 2022". However, I found the correlation between "research growth" and "introduction of CoT" claimed by the authors not convincing enough, as 2022 is the year when ChatGPT comes out, and everything related to LLM research should grow considerably since then. the author also does not show papers in 2020 and 2021 (since GPT-3 comes out in 2020, which marks the beginning of in-context learning and enables inference-time optimization) to show the existence of "acceleration", or show papers in other LLM areas to show that research on regimes and architectures are growing particularly fast. I would expect the authors to fix the statement in the caption of Fig. 1.

2. Some references are missing in this paper:

a) This paper did not mention Anthropic's Model Context Protocol (MCP), which is a prevalent work of current agent systems that tries to  build the infrastructure for future agentic models. I would suggest the authors to add discussion about this.

b) As ReAct (https://arxiv.org/abs/2210.03629, which I sugget the authors to cite) unifies agent acting with reasoning, LLM (single) agents that interacts with the environment can also utilize inference scaling of standalone LLM methods, such as exploration and search. This is also an area with many works, such as LATS (https://arxiv.org/abs/2310.04406) and RAFA (https://arxiv.org/abs/2309.17382).

c) The paper is missing one line of emerging works in Sec. 3.1.1: many-shot In-Context Learning (ICL; see https://arxiv.org/abs/2404.11018), which aims to increase the number of demonstrations to hundreds and thousands to fully utilize current LLM's growing context length. Also, the connection of "demonstration engineering" and ICL should be more clarified in this part.

3. The introduction of RL algorithms has several problems.

a) The definition of advantage between Eq. 3 and Eq. 4 is incorrect. The author states "the state value is equivalent to the expected value ... marginalized over all possible actions available". Advantage is always accompanied with a particular policy, and the expectation is not counted over all possible actions, but counted with weights being the probability of taking the action (https://spinningup.openai.com/en/latest/spinningup/rl_intro.html). For example, for a deterministic policy with one action of probability 1 over 10 possible actions, the Q-value is not counted for the remaining 9 actions with 0 probability. The introduction also does not properly connect $\pi_{\theta_o}$ and $\pi_{\theta_\text{ref}}$, as the latter is often not changing but the former is always changing ("from the previous episode or iteration").

b) REINFORCE is not necessarily a method that optimizes the weighted objective of the entire response (i.e. only has a single step for the whole episode; https://link.springer.com/article/10.1007/BF00992696).

c) Overall, I think the introduction of the high-level motivation of RLOO and GRPO (and other possible RL algorithms to add, such as DAPO (https://arxiv.org/abs/2503.14476)) could be improved. The current paper does not mention the reason why they are invented - to remove the need of critic in PPO and accelerate training, which is an important motivation for many RL algorithm variants for LLMs.


**Minor Changes**

1. In Sec. 6.1.1, P32, "... even MCTS Qin et al. (2024b)." should be "... even MCTS (Qin et al., 2024b)".

2. in Tab. 3 caption, the period is missing.

3. In Sec. 3.2.2, P17, "3Moreover" should be "Moreover".

4. In Eq. 5, $L_{REINFORCE}$ should be $L_\text{REINFORCE}$.

5. I would suggest to add a brief discussion of the verifier work mentioned in Sec. 6.1.3 to Sec. 4.2.

6. In Fig. 4, some texts in the boxes are centered while others are not; it will be better if the format can be unified.

**Strengths And Weaknesses:**

**Strengths**

1. The paper seems to be a well-rounded investigation into the related literature, which is clearly written, easy to follow, and includes many different aspects of LLM reasoning.

2. The paper points out the current trend and important future challenges of the LLM agent, which is a timely contribution to the LLM community.

**Weaknesses**

See requested changes.

---

> ### Author Response · Authors · 2025-05-24
> **Initial response**
>
> Thank you for recognizing the clarity of our writing, the timeliness of our contributions, and for appreciating our discussion of challenges and trends. Please see our response to your comments/suggestions below.
>
> **Major C1: In the caption of Fig. 1, I found the correlation between "research growth" and "introduction of CoT" claimed by the authors not convincing enough**
>
> Thank you for the suggestion. Actually, our statements were made considering the growth of research interests in “reasoning” (the main focus of this paper). But we agree that the growth is also influenced by many other factors, including the introduction of ChatGPT, popularity of in-context learning, instruction tuning, and chain-of-thought (CoT). We have revised the tone accordingly and acknowledge the potential influence of other factors.
>
> **Major C2.a:  This paper did not mention Anthropic's Model Context Protocol (MCP)**
>
> Thank you for the suggestion! We have added the discussion of merits and limitations of MCP as an open standard to connect models with real-world data sources in Section 3.2.2.
>
> **Major C2.b:  As ReAct unifies agent acting with reasoning, LLM (single) agents that interacts with the environment can also utilize inference scaling of standalone LLM methods, such as exploration and search. This is also an area with many works, such as LATS and RAFA**
>
> Thanks for the references! Actually, ReAct was included in our earlier version but was unfortunately commented out mistakenly in the editing process. These are important works on single-agent systems. We have incorporated discussions on ReAct, and follow-ups such as LATS and RAFA in Section 3.2.
>
> **Major C2.c: The paper is missing one line of emerging works in Sec. 3.1.1: many-shot In-Context Learning**
>
> Thank you for the helpful suggestion regarding emerging works on many-shot In-Context Learning (ICL)! We have updated Section 3.1.1 to include a line highlighting many-shot ICL with several many-shot ICL studies. Additionally, we have clarified the connection between "demonstration engineering" and many-shot ICL: many-shot prompting can be seen as an extreme form of demonstration engineering, where the focus is on scaling the quantity of demonstrations to maximize the model’s capacity to learn from in-context examples.
>
> **Major C3: The definition of advantage between Eq. 3 and Eq. 4 is incorrect. REINFORCE is not necessarily a method that optimizes the weighted objective of the entire response. Overall, I think the introduction of the high-level motivation of RLOO and GRPO (and other possible RL algorithms to add, such as DAPO could be improved.**
>
> Thanks for your valuable comments on the RL definitions. We agree with your notions and we updated the paper to clarify further. Indeed, the selection of actions (calculation of Q and V, and in turn advantage)  is conditioned on the weighted probabilities of the current policy. We also added extra information about RLOO, GRPO to the paper, including the removal of the value/critic model and better stability.
>
> **Minor C.5: add a brief discussion of the verifier work mentioned in Sec. 6.1.3 to Sec. 4.2.**
>
> Thank you for the suggestion, we have updated the discussion at the beginning of Section 4.2 to properly set expectations (highlighted in blue): Section 4.2 covers training methodologies, whereas Section 6.1.3 takes a broad view, framing verification itself as a reasoning problem and highlighting both concrete works and associated analyses.
>
> **Minor C.6: In Fig. 4, some texts in the boxes are centered while others are not; it will be better if the format can be unified.**
>
> Thank you for the suggestion and we have updated the figure 4 to make everything unified.

---

> > ### Comment · Reviewer_fmKu · 2025-05-25
> >
> > Thanks for the authors' response. I think they addressed my concern and thus I recommend acceptance of the paper.

---

> > > ### Author Response · Authors · 2025-05-26
> > > **Thanks!**
> > >
> > > Thanks for your recommendation!

---

### Review · Reviewer_bHdo · 2025-05-08

**Summary Of Contributions:**

This work presents a comprehensive survey of reasoning in large language models (LLMs). It organizes existing research along two primary dimensions: (1) the regime, which distinguishes between reasoning achieved through training versus inference-time methods, and (2) the architecture, which differentiates between standalone LLMs and agentic systems. Additionally, the survey provides an overview of various learning algorithms employed in the development of reasoning-capable systems. Finally, it highlights emerging trends and outlines key open challenges in the field.

**Audience:**

Yes

**Claims And Evidence:**

Yes

**Requested Changes:**

- I think the terms “regime” and “architecture” , which are the main 2 dimensions of categorization in this paper, are not well-chosen. Architecture is more commonly used to refer to different neural architectures while the term regime can have multiple interpretations. Thus, while the 2 dimensions make sense, their naming can be improved.
- In the comparison to related surveys subsection, it will be useful to more concretely provide details about how this survey differs from existing ones. This can help direct the reader to the right survey.
- What is the exact difference between a standalone LLM and a single-agent system? For example, if I use a best-of-N inference algorithm using a reward model, it seems to satisfy both the single-agent system and the standalone LLM from Figure 2.  Specifically, standalone LLM has an aggregation step shown in Figure 1, I guess this aggregation step has to be very constrained and cannot use any external tools? Similarly, distillation is a part of both Sections 5.1 (Learning to Reason with Standalone LLM) and Section 5.2 ( Learning to Reason with Single-agent Systems).
- Section 5.1.1 says “Sampling high-quality trajectories begins with high-quality prompts”. While this step might be a part of some pipelines, I don’t think  high quality prompts are a requirement to obtain high-quality trajectories (for eg., the base model might already output high quality trajectories). The statement should be refined.

**Strengths And Weaknesses:**

Strengths-

- The survey is well-written and easy to follow. It provides necessary background where needed, making it easier for a broader audience to follow while maintaining low-level technical detail.
-  The dimensions chosen for categorization of existing work is interesting and present the state of the field in a new light.
- The survey is comprehensive and covers a significant amount of literature in the area.

Weaknesses-

- There are numerous surveys in this area and the work does not concretely specify how it is different from them.

---

> ### Author Response · Authors · 2025-05-24
> **Initial response**
>
> Thank you for recognizing the clarity of our writing, the novelty of our choice of dimension for categorization and the comprehensiveness of our paper! Please see our response to your comments/suggestions below.
>
> **W and C2: more concretely provide details about how this survey differs from existing ones**
>
> Thank you for raising this point! Although we tried our best to distinguish our work from other concurrent surveys in “Sec 1.5: Comparison to Related Surveys”, we agree this can be further improved. Thus, in the revised manuscript, we have clarified further (see texts in blue) how our survey differs from prior work by adding explicit comparisons in Sec 1.5. We provide more concrete discussions of prior surveys, which often focus on either the development of techniques (e.g., in-context learning, inference scaling, reinforcement learning for LLM reasoning) or regimes (LLMs, single or multi-agent systems) in isolation, whereas our work provides a unified perspective by organizing techniques under different regimes. We highlight how these techniques evolve and interact within each setting. These changes are intended to help readers better understand our unique contributions and navigate related literature more effectively.
>
> **C1: while the 2 dimensions “regime” and “architecture” make sense, their naming can be improved**
>
> Thanks for your suggestion. We will consider changing “regime” to “paradigm” and  “architecture” to “system design” to avoid confusion, if reviewers think these are better.
>
> **C3: Difference between a standalone LLM and a single-agent system?**
> **(a) if I use a best-of-N inference algorithm using a reward model, it seems to satisfy both the single-agent system and the standalone LLM from Figure 2**
> **(b) standalone LLM has an aggregation step shown in Figure 1, I guess this aggregation step has to be very constrained and cannot use any external tools?**
> **(c) distillation is a part of both Sections 5.1 (Learning to Reason with Standalone LLM) and Section 5.2 ( Learning to Reason with Single-agent Systems)**
>
> Thanks for the question, these are subtle but important concepts. We are happy to clarify:
>
> The main distinction is the interactiveness and autonomy (Section 2.3.2). More specifically, we define “a system with agent-environment interaction as a single-agent system” (last paragraph in Section 2.3.2) and we define the environment as “an external module that the agent cannot modify” (second last paragraph Section 2.3.2). Under this definition,
> - For question (a). A reward model is considered part of the environment, since it is external and not modifiable by the agent. Therefore, such a setup is categorized as a single-agent system in our framework.
> - For question (b). Yes, the aggregation in standalone LLM is constrained and cannot use external tools, as tools are actions to interact with the environment (and will receive some feedback from the environment). An example of aggregation in standalone LLM is Self-Consistent CoT.
> - For question (c). Yes, distillation as a technique indeed can be used in both architectures. We believe that such intersections are valuable and could inform future developments.
>
> Figure 2 is intended to provide a high-level view of our categorization. For a more detailed breakdown, we recommend referring to Figure 5, which shows more concrete distinctions. We also want to note that agenticness is a spectrum and definitions could be varied across work. We use above definitions to better serve our categorization, by no means this is exclusive with other potential definitions.
>
> **C4: Section 5.1.1: “Sampling high-quality trajectories begins with high-quality prompts”. While this step might be a part of some pipelines, I don’t think high quality prompts are a requirement to obtain high-quality trajectories (for eg., the base model might already output high quality trajectories).**
>
> We agree our previous description has caused some confusion. What we want to emphasize here is that we need diverse prompts with high quality to cover the knowledge space for knowledge distillation or mode seeking. One evidence is that: although DeepSeek-V3 is the strongest base model, we can hardly train another DeepSeek-R1 by sampling trajectories using Deepseek-V3 for reinforcement learning while using only the prompts from GSM8K. The core insight behind this is high quality prompts can introduce more general exploration space, which would both benefit distillation and reinforcement learning. We have refined the description accordingly.

---

### Review · Reviewer_zxXm · 2025-05-11

**Summary Of Contributions:**

The paper provides a survey of works on using LLMs for reasoning from the advent of Chain of Thought in 2022 to Deepseek R1 in late 2024. On the way it covers works on augmenting both LLM inference (inference scaling) and training (learning to reason) for reasoning. It considered approaches using standalone LLMs, single agent systems and multi agent systems. The paper concludes with a discussion of observed trends in research on LLM reasoning and open challenges around evaluation, interpretability, and data efficiency.

**Audience:**

Yes

**Claims And Evidence:**

Yes

**Requested Changes:**

It is understandably difficult to summarize the vast number of works that have appeared in this space in such a short span of time and balance breadth and depth. I would recommend explaining the first/most important method in each Section or subsection in a bit more detail with examples/figured and then listing the rest of the works as they currently are. That way the more detailed explanation for 1-2 works would give more context for the rest of the works.

The following are some other things that were not entirely clear to me:

1. Figure 6 is not entirely clear. Why are there inference scaling and reasoning step nodes at the same level? How is this connected to learning to reason? It might be better to have separate figures for inference scaling and learning to reason.

2. Under Demonstration Engineering in 3.1.1 it is mentioned that exemplar quality is the limiting factor rather than exemplar relevance but the difference between the two was not clear to me. I feel an example would be helpful in illustrating the difference.

3. What do you mean by "cooperative nature" of LLMs in 3.3.2?

4. How is the Markov property satisfied if $s_{t+1} = [s_t ; a_t]$ ? Won't s_t contain all previous states anyway?

5. Can you give an example of how the annotation for a reasoning step can be obtained by rolling out the response in MCTS (4.2.2) instead of using human annotation?

6. What kind of resources are dynamically allocated based on prompt complexity in cost-aware reasoning (5.4.1)?.

7. Why is BoN sampling misaligned with fine-tuning objectives?

**Strengths And Weaknesses:**

The main strength of the paper is the coherent categorization and organization of the vast literature in this space. The classification of works into learning to reason and inference scaling cleanly separate research on training and inference respectively. Likewise the clarification of works into standalone LLM, single agent, and multi-agent systems helps to clearly understand and appreciate the additional complexities as agents are incorporated in reasoning pipelines.

The main weakness of the paper is the limited exposure it provides to some key ideas. Due to the large number of papers in many areas, some subsections end up feeling like an enumeration of papers and do not clearly highlight the key technical ideas.

---

> ### Author Response · Authors · 2025-05-24
> **Initial response (1/2)**
>
> Thank you for recognizing the coherence of our categorization and appreciating our two dimensions: learning-to-reason vs. inference-scaling, and standalone LLMs vs. single-agent systems vs. multi-agent systems. Please see our response to your comments/suggestions below.
>
> **W:  limited exposure to some key ideas**
> Thanks for the suggestions! Although we tried our best to do that in our first version, we agree that some sections can be further improved. We thus updated the draft as follows:
>
> - **In Section 3.1.1**, we have added the main motivations and core ideas at the beginning, and provide a more detailed discussion of key works (highlighted in blue).
> - **In Sections 3.1.2, 3.2, and 3.3**, we presented the content to follow a coherent structure, where subsections start by introducing the main motivation or core ideas.  Representative key works are illustrated in further details. The remaining works are discussed more briefly to provide a broader overview while maintaining focus on the main conceptual trajectory.
> - **In Section 4**: We have revised this section to better motivate key reinforcement learning methods such as RLOO and GRPO, with additional explanations (highlighted in blue in Section 4).
> -  **In Section 5**: For each subsection, whenever possible, we begin with high-level motivation, discuss the strengths and limitations (if any), introduce representative work in 1–2 sentences, and then briefly mention follow-up studies. For example, in Section 5.1.3 on training with latent reasoning, we first explain what latent reasoning is and why it is important, then describe [1] in detail, followed by a brief discussion of other related work.
>
> Additionally, we include **Tables 2, 3, 4, 6, and 7** as high-level summaries (covering perspectives, methods, characteristics, and representative works) for the subsections in Sections 3 and 5. We hope these tables help clarify the structure and key ideas.
>
> We will be happy to revise the paper further if you have any further concerns.
>
> **C1: Figure 6 is not entirely clear**
>
> Thanks and there might have been a misunderstanding:
>
> - Inference-scaling is not placed at the same level as reasoning steps. As noted in the caption, the star-shaped node indicates a step **selected by** an inference-scaling method (e.g., using an external verifier, assuming PRM). The purpose is to illustrate that inference-scaling involves sampling multiple candidate trajectories and then selecting one based on an additional mechanism.
> - We intentionally included inference-scaling and learning-to-reason in the same figure to highlight their connection. Specifically, offline learning-to-reason methods rely on preference trajectories that often come from **the same** sampling processes as inference-scaling. To reflect this, we reused the sampling illustration but added preference labels at the end of each trajectory to indicate the preference data. We also revised the caption and visual elements of the figure to make this distinction clearer.
>
> **C2: Sec. 3.1.1. the difference between exemplar quality and relevance was not clear**
>
> Thank you for raising this! We have clarified this point in the revision. As highlighted by [2], LLMs often fail at analogical reasoning not because the exemplars are topically irrelevant, but because the exemplars, regardless of relevance, may be factually incorrect or structurally unhelpful. For example, biological examples can still improve math performance on GSM8K if they are accurate and structurally aligned with the target problem, as extensively experimented by [2]. This illustrates that quality i.e., correctness, clarity, and utility for reasoning, is more critical than relevance alone. We've updated the text subsection 3.1.1 to include this detailed discussion to better distinguish the two concepts.
>
> **C3: What is "cooperative nature" of LLMs in Sec.3.3.2?**
>
> Here we refer to the ability of LLMs to interact constructively and effectively with one another, working together as if they were a team of agents with a shared goal. To avoid ambiguity, we will rephrase it to 'collective strength' which is a more accurate description.
>
> **C4: How is the Markov property satisfied if Won't s_t contain all previous states anyway?**
>
> This is a subtle but important point, and we are happy to clarify.
> $s_{t+1} = [s_t;a_t]$ does satisfy Markov property. For the autoregressive language modeling task, the next token depends on all the previous ones. Therefore, to make LM task "markovian" for RL to work, we define the "current state" as all previous tokens, such that it no longer needs to know what happened that led to those tokens. We have added this explanation in Section 4 (highlighted in blue) to improve clarity.
>
> [1]: Implicit chain of thought reasoning via knowledge distillation. Deng et al., 2023.
> [2]: Relevant or Random: Can LLMs Truly Perform Analogical Reasoning? Qin et al., 2025

---

> > ### Author Response · Authors · 2025-05-24
> > **Initial response (2/2)**
> >
> > **C5: Example of how the annotation for a reasoning step can be obtained by rolling out the response in MCTS (4.2.2) instead of using human annotation?**
> >
> > Yes, For example, by rolling 5 completions randomly from the same prefix $y_{\leq t}$, and supposing that 3 of them have reached the correct answers, the confidence that the prefix $y_{\leq t}$ is correct can be approximated as 0.6. The coarse signals can thus be used to train a pseudo PRM, these methods have been used in [3,4]
> >
> > We have added this explanation to Section 4.2.2 (highlighted in blue) to improve clarity.
> >
> > **C6: What kind of resources are dynamically allocated based on prompt complexity in cost-aware reasoning (5.4.1)?.**
> >
> > In most cases, the resources refer to "computation resources", for example, increasing the sampling times in parallel or employing reasoning models for longer and deeper reasoning trajectory.
> >
> > **C7: Why is BoN sampling misaligned with fine-tuning objectives?**
> >
> > For the perspective of distribution, Best-of-N distribution defines the probability that a random generation $y’$ is strictly worse than y, i.e., $p_{\le}(y) = \Pr_{y'\sim \pi_{\mathrm{ref}}}\bigl[r(y') \le r(y)\bigr]$. In both standard SFT (or RFT) and reinforcement learning, usually we do not assume the target $y$ has the highest reward, which causes misalignment.
> >
> > [3] Math-Shepherd: Verify and Reinforce LLMs Step-by-step without Human Annotations. Wang et al., 2023.
> > [4] Learning Planning-based Reasoning by Trajectories Collection and Process Reward Synthesizing. Jiao et al., 2024.

---

### Decision · Action_Editor_TBEb · 2025-06-25

**Recommendation:** Accept as is

**Additional Comments:**

This is a timely and well-written survey paper. The two-axis breakdown of inference scaling vs. architecture of the system is an interesting one and is a nice perspective. This categorization and the overall writing resonated with all of the reviewers, and all of them ended up positive on the work. As with many surveys, the chief drawbacks of the paper are (a) inability to cover everything, especially newer content (e.g., MCP is now widely known and used, whereas in March when the survey was submitted, it was less widely known); (b) the survey looking like a laundry list of topics.  However, this paper does a reasonably good job of navigating these issues and the update from the authors helps it further.

**Audience:**

Yes

**Audience Explanation:**

Yes, this is a highly-studied topic and a survey on such a topic would likely be read.

**Claims And Evidence:**

Yes

**Claims Explanation:**

This paper claims to provide a categorization and analysis of existing methods, appropriate for a survey paper.